# Learning Mathematical Rules with Large Language Models

**Antoine Gorceix**[*]
J.P. Morgan AI Research

**Bastien Le Chenadec**[*]
J.P. Morgan AI Research

**Ahmad Rammal**[*]
J.P. Morgan AI Research

**Nelson Vadori**
J.P. Morgan AI Research

**Manuela Veloso**
J.P. Morgan AI Research

## Abstract

In this paper, we study the ability of large language models to learn specific mathematical rules such as distributivity or simplifying equations. We present an empirical analysis of their ability to generalize these rules, as well as to reuse them in the context of word problems. For this purpose, we provide a rigorous methodology to build synthetic data incorporating such rules, and perform fine-tuning of large language models on such data. Our experiments show that our model can learn and generalize these rules to some extent, as well as suitably reuse them in the context of word problems.

## 1 Introduction

We focus on a specific aspect of mathematical reasoning, namely the ability of large language models (LLMs) to learn specific abstract mathematical rules and to reuse them while answering word problems, namely questions formulated in natural language that are usually made up of a few sentences describing a scenario that needs to be solved through mathematics. We will also refer to these mathematical rules as "skills". Thus, we study the following research question: *Can LLMs learn and generalize specific mathematical rules, and apply them in contexts that have not been seen during training, for example when answering word problems?*

We will fine-tune models on carefully built synthetic data reflecting the mathematical rules of interest, and we provide a detailed description of our methodology for building such data in section 2 and appendix B. Broadly speaking, we want our training data to be presented similarly to what you would find in a mathematics textbook. For example, focusing on how to rearrange or simplify an equation, or how to use the distributivity property of the addition operator. But *without* word problems.

The data at test time, however, will contain word problems where the model first needs to translate the question in natural language into one or more equations, and then use various rules to solve these equations. This is what we will call ***bottom-up generalization***, namely the model's ability to go from mathematical rules to answering word problems. We also provide an empirical study on the model's ability to generalize when increasing the mathematical complexity of the task. That is, when we increase parameters such as the number of variables, the number of equations, the variable name token length, and so on. We will call this ***top-down generalization***, as we zoom into a given mathematical rule to make it more complex. We comment on the related work in appendix A.

**Our contributions.** We provide a rigorous methodology to create synthetic data containing specific mathematical rules such as manipulating equations (section 2). We show how fine-tuning models on those rules allows them to be reused in the context of word problems, while maintaining performance

---

[*]These authors contributed equally to this work.

38th Conference on Neural Information Processing Systems (NeurIPS 2024).

on usual benchmarks (section 3). We conduct experiments showing that models can generalize these rules to some extent when we increase the mathematical complexity of the problem, such as the number of variables (section 4). We find that allowing variables to take value in a larger set of tokens improves the ability of the model to generalize distributivity to unseen variable names. Finally, we provide an algorithm to extract mathematical expressions from text data detailed in appendix E, which we use to evaluate all models.

## 2 Building Synthetic Data Incorporating Mathematical Rules

We aim at constructing synthetic data reflecting specific mathematical rules that we would like our model to learn. The detailed description of the synthetic data is provided in appendix B, and is presented similarly to what you would find in a mathematics textbook, *without* word problems.

In section 3 we focus on bottom-up generalization. To this extent we consider the following rules presented in section B.1: a) finding the roots of a quadratic polynomial; b) solving for a given variable in a linear equation in that variable, for example: *solve for the variable $x$ in $13 \cdot x \cdot y + 24 \cdot y^2 = 17 \cdot x \cdot z + 12 \cdot y \cdot z$. $x = \frac{12 \cdot y \cdot (z - 2 \cdot y)}{13 \cdot y - 17 \cdot z}$*; c) simplifying terms in an equation, for example: *simplify the following expression: $-9 \cdot x^2 - 8 \cdot y + 5 \cdot y$. By grouping terms: $-9 \cdot x^2 - 3 \cdot y$*; d) isolating a variable in an equation of the form $a = b * (c + d + \dots)$ or $a = b/(1/c + 1/d + \dots)$.

In section 4 we focus on top-down generalization. We consider rules such as distributivity, exponentiation, manipulation of single and pairs of equations as well as solving single steps of Gaussian elimination, all presented in section B.2. In particular, we focus on the model's ability to apply those rules when the variables appearing in the equations are arbitrary strings, and study the impact on generalization performance. For example: *Q: Expand this expression: $(-5 + soccer - dog) * (blue - sky)$. A: By the distributivity property: $-5 * blue + 5 * sky + soccer * blue - soccer * sky - dog * blue + dog * sky$.* The goal is to learn the fundamental nature of the corresponding operators, which do not depend on the variable names.

## 3 Bottom-Up Generalization: Going from Mathematical Rules to Word Problems

We consider three classes of word problems. Finding a solution to these problems involves mainly two steps. First, the model needs to correctly translate the question in natural language into one or multiple equations. Second, the model needs to apply one or more mathematical skills to correctly manipulate the equations to answer the question. We choose our examples such that the baseline model is able to perform the first step but struggles on the second. Our goal when performing fine-tuning is as follows: keep the general knowledge required to solve the first step, while providing new tools to also solve the second. We write the solutions to these problems using chain-of-thought, i.e. we break down the reasoning in several steps. We will fine-tune Llama-3 8B Instruct on on a mix of synthetic data described in section B.1 together with data from the Orca dataset Mukherjee et al. (2023) acting as a regularizer in order to avoid catastrophic forgetting and maintain performance on common benchmarks (see section C.4). Additional details are provided in appendix C.

**Quadratic Polynomials.** We consider simple geometric problems involving the calculation of areas for shapes with the same unknown side length. The total area of all shapes is known, while the unknown side length is to be determined. The solution involves two main steps. The first one is to formulate a quadratic equation that models the relationship between the known areas and the unknown side length. This equation is always a quadratic polynomial. The second step is to find the roots of this polynomial to find the unknown side length. Problems are constructed to always yield one positive and one negative root, ensuring no ambiguity in the solution. We generate many such problems by varying the names (pools, fields, etc.) and shapes (square, triangle, parallelogram, rectangle) of the surfaces, the number of such surfaces per shape (in average, 2.5 surfaces per shape per word problem), as well as the values of the areas and side lengths appearing in the problem. We present an example of prompt given to the model in Figure 1.

We provide three examples of prompts and responses to the model (3-shot), and assess the validity of the numerical solution. We evaluate the performance of our fine-tuned model against a baseline across varying levels of difficulty. The difficulty scale ranges from 0 to 100%, in increments of 20%,

corresponding to the proportion of problems with non-integer roots. Non-integer roots present a greater challenge as they require more complex calculations compared to integer roots, which the baseline model often guesses via straightforward factorization. To accurately determine non-integer roots, the discriminant of the polynomial must be calculated. We present our results in table 1.

While the baseline model successfully translates the problem into an equation, it encounters difficulties in finding the roots of the quadratic polynomial. Instead of applying the discriminant method, it prematurely attempts a hazardous factorization, often leading to errors. In contrast, the fine-tuned model consistently utilizes the discriminant approach, a result of targeted fine-tuning on a quadratic polynomial solving task. As a consequence, our fine-tuned model is able to outperform the baseline.

Figure 1: Word problem example - quadratic polynomial.

**Prompt**: Jim has a total of 2 swimming pools. The first swimming pool is a square, with an unknown side length x. The second is a rectangle with one side measuring 1 meters and the other being the unknown side length x. The total area covered by these swimming pools is 2 square meters. What is the unknown side length x?

Table 1: Accuracy (%) - quadratic polynomials (3-shot).

| Difficulty | 0 | 20 | 40 | 60 | 80 | 100 |
|---|---|---|---|---|---|---|
| **Llama-3 8B Instruct** | 14 | 10 | 10 | 10 | 9 | 8 |
| **Llama-3 8B Fine-tuned** | 39 | 39 | 37 | 36 | 35 | 35 |

**Physics Problems Involving Resistors.** We consider simple electrical circuits composed of $n$ resistors connected either all in parallel or all in series. The model is tasked with finding the equation governing the circuit and isolating a variable in the equation. We provide 3 few-shot examples of prompts and answers to the model, and then evaluate the correctness of the symbolic solution. We present an example of prompt given to the model in Figure 2. We compare the performance of a

Figure 2: Word problem example - resistor circuit.

**Prompt**: You have a circuit with the following resistors: [R1 ∥ R2]. Given that the current flowing through the circuit is I amp and the voltage across the circuit is U volts, express the resistance of R1 in terms of the other variables.

fine-tuned model against a baseline model on different configurations of $n$ resistors either in parallel or in series. For each configuration (i.e. [R1 ∥ R2], [R1 - R2], etc.), 100 examples are generated by first generating a prompt for each possible unknown, and then sampling multiple responses from the model if there are not enough unique problems (using a temperature of $0.1$ and top-5 sampling). The fine-tuning data only contains variables from the *restricted vocabulary* (cf section B), with the skill of isolating a variable in an equation of the form $a = b * (c + d + \dots)$ or $a = b/(1/c + 1/d + \dots)$. Table 2 shows the results of the models on resistors in series and parallel. The fine-tuned model performs perfectly on resistors in series. On resistors in parallel, the fine-tuned model performs significantly better than the baseline model, but still struggles with more complex configurations despite being trained on these equations.

**Fruit Baskets.** Alice and Bob buy different quantities of fruits, each with a specific price, and both end up paying the same total amount. The goal is to find the price of one fruit based on the prices and quantities of the others. We generate these problems by varying the names of the fruits, the number of fruits, and the relationships between their quantities and prices. In our examples, all the fruits depend on the quantities and price of the first fruit, as illustrated in Figure 3. This involves setting up an equation, substituting variables, simplifying it, and solving for the unknown. The problem reduces to solving a linear equation involving three variables. We can increase the complexity of the problem by simply increasing the number of fruits that are purchased.

We evaluate the model's answer by extracting its symbolic output and comparing it to the correct one. We observed an improvement in the fine-tuned model compared to the Llama-3 8B Instruct

Table 2: Accuracy (%) - resistors problems of increasing complexity (3-shot).

| Number of resistors | Resistors in series | | | | Resistors in parallel | | | |
|---|---|---|---|---|---|---|---|---|
| | 2 | 3 | 4 | 5 | 2 | 3 | 4 | 5 |
| **Llama-3 8B Instruct** | 98 | 100 | 81 | 100 | 58 | 8 | 14 | 17 |
| **Llama-3 8B Fine-tuned** | 100 | 100 | 100 | 100 | 68 | 73 | 60 | 35 |

Figure 3: Word problem example - fruit baskets.

**Prompt**: Alice and Bob went to the grocery store and bought the following items:
- bananas: Alice bought $q_{A_1}$, and Bob bought $q_{B_1}$. The price of a single one is $p_1$.
- blueberries: Alice bought $q_{A_2}$ where $q_{A_2} = 2 \times p_1$, and Bob bought $q_{B_2}$ where $q_{B_2} = 8 \times q_{A_1}$. The price of a single one is $p_2$ where $p_2 = 5 \times q_{B_1}$.
Both ended up paying the same total price. Find the price of bananas in terms of $q_{A_1}$ and $q_{B_1}$.

baseline. The latter scores 19% versus 35% with the former in the case of two fruits. For three fruits, the baseline scores 0% against 13% for the fine-tuned model. While the baseline model can set up the full equation, it struggles with simplifying or solving it. The fine-tuned model, though not perfect, performs better in finding the correct answer due to its fine-tuning on mathematical tasks such as expression simplification and equation solving.

We also considered assigning numerical values to the variables in that problem, requiring the model to perform calculations to find a numerical solution instead of a symbolic one. We illustrate this in figure 9. We found that the models' performances were very similar to the symbolic case. We refer to the appendix C.3, table 3 for a full overview on the results.

**On the necessity to align training data with the word problems.** Our experiments reveal limitations in the ability to generalize specific mathematical rules to word problems. First, we have to make sure the form of the equations in the training data is the same as the form of the equations in the test data. For instance, the model is unable to perform on the resistors in parallel problem when it is trained to put fractions over a common denominator in its answer. We say that the model is **triggered** in the wrong modality. Another example with the resistors in parallel is as follows: if in the training data, we put equations in the form "$A/B$", and in the test data we put them in the form "$A * 1/B$", the model will not be able to perform. We say that the model is not **triggered**. This second example illustrates the importance of the choice of the few-shot examples, because the form of the equation will generally match the form in the few-shot examples.

## 4   Top-Down Generalization: Increasing the Mathematical Complexity of the Task

We fine-tune Llama-2 7B Chat on *all* the synthetic data described in section B.2 (and that data only) and evaluate the model's ability to perform the following rules: *distributivity, commutativity, division, exponentiation, variable evaluation, remarkable identities, single equation and two equations manipulation*. We check in section D.4 that the performance on general knowledge benchmarks remains relatively stable, and present experimental details in section D.6.

**Solving systems of equations by recursive call of our model**. In section D.2, we train the model on individual steps of the Gaussian elimination algorithm and show at test time that we can solve a full system by recursively calling the model. An example is presented in figure 5.

**Experiments on various mathematical rules**. We analyze the ability of our model to perform the rules of interest under different configuration regarding the training vocabulary. We find that our fine-tuned Llama-2 model consistently outperforms other models across all considered mathematical rules. The full results are presented in section D.3. In section D.5, we present the model's performance on some word problems, in particular we emphasize its ability to ***combine skills*** in figures 15 and 16.

**Ablation study on the distributivity rule**. In section D.1 we focus on the distributivity rule. We train the model on data where the variables appearing in the equations take value in subsets of increasing

tokenizer vocabulary sizes. We find that training on larger vocabulary sizes improves the ability of the model to generalize distributivity to unseen variable names as well as to increasing the number of variables, see figure 4.

Figure 4: Validation accuracy on the distributivity rule for different vocabulary sizes. Each model is evaluated on the $(100 - x)\%$ complement of its training vocabulary $x\%$ (% of tokenizer's vocabulary). The dashed lines delimit the parameters seen during training from those unseen. From left to right, from top to bottom: $x = 1, 10, 50, 75, 95$.

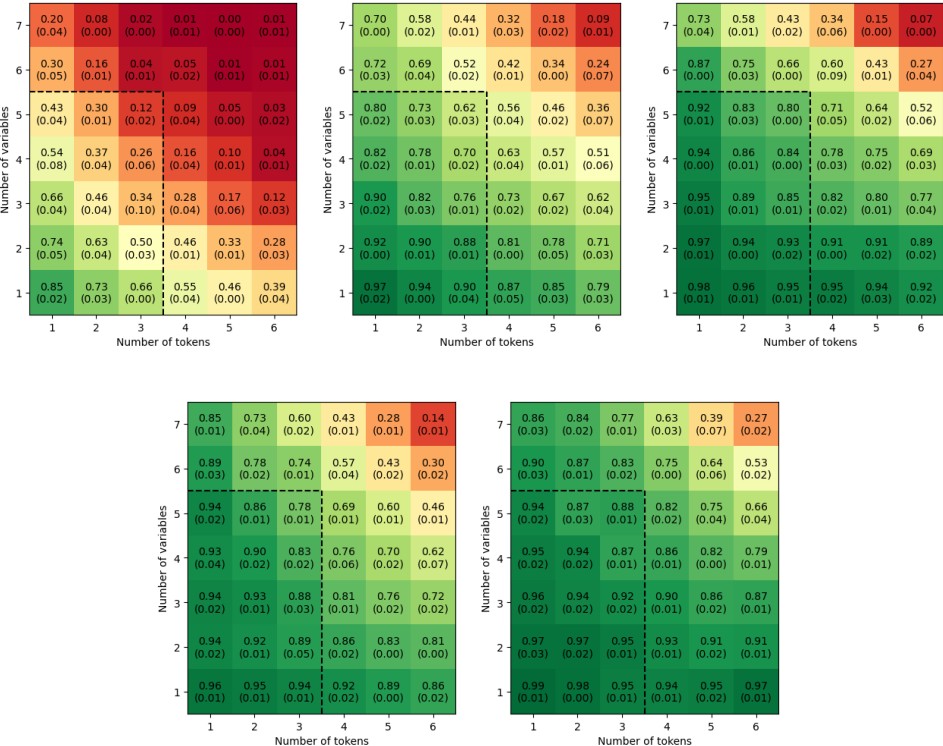

Figure 5: Detailed resolution of a system of equations by recursive call of our model on each step $i \rightarrow i + 1$. The variables are $dog, sky$.

$$\begin{cases} -9 * dog - cat * sky = -blueberry \\ 3 * dog - tree * sky = -6 \end{cases}$$

step 0

$$\begin{cases} dog + (cat/9) * sky = (blueberry/9) \\ 3 * dog - tree * sky = -6 \end{cases}$$

step 1

$$\begin{cases} dog + (cat/9) * sky = (blueberry/9) \\ -(tree + (cat/9) * 3) * sky = -(6 + ((blueberry/9) * 3)) \end{cases}$$

step 2

$$\begin{cases} dog + (cat/9) * sky = (blueberry/9) \\ sky = \frac{(6 + ((blueberry/9) * 3))}{(tree + (cat/9) * 3)} \end{cases}$$

step 3

$$\begin{cases} dog = ((blueberry/9) - (\frac{(6 - ((f/9) * 3))}{(tree + (cat/9) * 3)} * (cat/9))) \\ sky = \frac{(6 + ((blueberry/9) * 3))}{(tree + (cat/9) * 3)} \end{cases}$$

step 4

## 5    Conclusion

We showed how fine-tuning models on some specific mathematical rules allows them to be reused in the context of word problems (bottom-up generalization), and also focused on the ability to generalize specific rules such as distributivity and manipulating equations (top-down generalization). Future research directions could include building a robust and rigorous methodology to create word problem data from a set of mathematical rules.

## Disclaimer

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

# A    Related Work

In recent years, researchers have trained large language models (LLMs) on data of unprecedented size, and studying the emergent capabilities of these models on various tasks has been a central focus Bubeck et al. (2023). Nevertheless, mathematical reasoning remains a challenge, although the rate of improvement of LLMs on solving mathematical problems has been significant Ahn et al. (2024). A prize was even launched with the goal of getting AI to perform at gold medal level at the International Mathematical Olympiad XTX Investments (2024), emphasizing the importance of this research topic. In 2024, Numina won the first progress prize. Their recipe involved tool integration as well as fine-tuning DeepSeek Math Shao et al. (2024), a model built using scalable math pre-training.

Properly evaluating LLMs on mathematical reasoning tasks presents significant challenges, among which: i) data contamination: since LLMs are trained on vast amounts of data, it is often not clear that some problems have not been seen during training; ii) assessment of proof correctness: it is not easy to automatically determine whether a sequence of mathematical reasoning steps - and more generally a proof - is correct. On this topic we note the work conducted with proof assistants such as Lean Yang et al. (2023), for which we can immediately determine whether a proof is true or false as mathematical reasoning is seen as a computer program; iii) the variety of mathematical problems is significant and their complexity/difficulty is not trivial to establish. On this topic we believe that work allowing to suitably categorize problems would be beneficial to the community.

ToRA Gou et al. (2024) combines natural language reasoning (rationale) with symbolic solvers (programmatic reasoning). WizardMath Luo et al. (2023) applies Reinforcement Learning from Evol-Instruct Feedback (RLEIF) to the domain of mathematics. MathPrompter Imani et al. (2023) uses the zero-shot chain-of-thought prompting technique to generate multiple algebraic expressions to solve the same problem in different ways and thereby raises the confidence level in the output results. Alphageometry Trinh et al. (2024) significantly advances the state-of-the-art on geometry problems at the level of International Mathematical Olympiads. Llemma Azerbayeva et al. (2024) is a suite of models tailored for mathematical reasoning. Galactica Taylor et al. (2022) is a large language model that can store, combine and reason about scientific knowledge. Minerva Lewkowycz et al. (2022) solves scientific and mathematical questions in natural language, generates step-by-step solutions using Latex notation, and was trained on data composed of scientific and mathematical information. Yuan et al. (2023) applies rejection sampling fine-tuning (RFT), that uses supervised models to generate and collect correct reasoning paths in order to augment fine-tuning datasets. Zheng et al. (2023) proposes progressive-hint prompting that enables automatic multiple interactions between users and LLMs by using previously generated answers as hints to progressively guide towards correct answers. Wang et al. (2023) proposes plan-and-solve prompting, that consists of two components: first, devising a plan to divide the entire task into smaller subtasks, and then carrying out the subtasks according to the plan. Bubeck et al. (2023) present among other things an experimental study of GPT4's mathematical reasoning capabilities. Jelassi et al. (2023) examine how transformers cope with two challenges: learning basic integer arithmetic, and generalizing to longer sequences than seen during training . Lample and Charton (2020) present a deep learning approach for tasks such as symbolic integration and solving differential equations. LeanDojo Yang et al. (2023) explores mathematical reasoning using the Lean proof assistant, among other things they suitably retrieve relevant premises from a vast math library. Mirzadeh et al. (2024) presents an interesting study on how model performance varies when slightly varying problems from the GSM8K benchmark. Chen et al. (2024) studies the impact of mixing different types of data on mathematical reasoning ability. Finally, the recent survey paper Ahn et al. (2024) presents the state of the field on the topic of mathematical reasoning.

# B Detailed Description of the Synthetic Data Incorporating Mathematical Rules

Our data is a combination of text and mathematical expressions, which are built from a set of *types* and *operations*. A *type* is a fundamental object that can be manipulated (a variable, an integer, a decimal number, etc.). By applying arithmetic *operations* to these types, we can construct complex mathematical expressions in the form of Abstract Syntax Trees (AST).

Specifically we consider the following types for our synthetic data:

- integers,
- decimals with up to two decimal places,
- arbitrary strings, either made from the concatenation of $k$ tokens taken from a subset of the tokenizer's vocabulary [1] which we call **full vocabulary**, or from the concatenation of a latin or greek letter with a digit which we call **restricted vocabulary**,

We overload our programming language's arithmetic operators $(+, -, *, /, \char`^)$ to implement a set of simplification rules. For instance, when adding two integers, we generally simplify the expression by evaluating the sum (though each simplification rule can be turned off). When no simplification applies, a node is added to the AST. This custom data structure allows us to control precise simplification details such as the order of terms, whether to evaluate numerical expressions, etc. In contrast, using a symbolic mathematics library such as Sympy would not allow us to control these details, as some simplifications are performed systematically. We also implement a translation function that converts our custom data structure to Sympy's data structure, and vice versa, to handle more advanced simplification rules externally, striking a balance between control and flexibility.

Types only hold their own value (e.g. 4, "x", 12.51), and the negative sign is considered a unary operation, while the other operations are binary. Thus our expressions are ASTs where each leaf is a type, and each internal node is a unary or binary operation. In simpler cases we might consider expressions of the form $\omega_1 \odot_1 \omega_2 \odot_2 \cdots \odot_{n-1} \omega_n$, where each $\omega_i$ is a type, and each $\odot_i$ is an arithmetic operation.

We are then able to represent our data structure as a string, taking into account operator precedence and associativity. We can thus construct mathematical expressions programatically, and generate prompts for our models. All of our prompts are of the form $*_{start}$ *Question* $*_{mid}$ *Answer* $*_{end}$, where $*_{start}$, $*_{mid}$ and $*_{end}$ are model-specific strings that separate the question from the answer. For example $*_{start} =$[INST], $*_{mid} =$[/INST] and $*_{end}$ is empty in the case of Llama-2 Chat. Similar string are used in the case of Llama-3 Instruct. In the below, $\omega_i$ and $\odot_i$ will denote respectively types and operations, which value can change depending on the context. For each one of the cases below, the number of terms appearing in the expressions is sampled randomly between some pre-specified lower and upper bounds.

## B.1 Rules Used in Section 3

For the rules below, when relevant we use Sympy to solve equations symbolically and factorize the result.

**Quadratic polynomials.** We want our model to solve quadratic polynomials. $\omega_1, \omega_2, \omega_3$, are random integers and $\omega_4$ is an arbitrary token.

- Question. *Solve this quadratic equation:* $\omega_1 \times \omega_4^2 + \omega_2 \times \omega_4 + \omega_3 = 0$.
- Answer. We first calculate the discriminant $\Delta = \omega_2^2 - 4\omega_1\omega_3$.
  - If $\Delta > 0$: The equation has two distinct real roots:
    * $\omega_4 = \frac{-\omega_2 \pm \sqrt{\Delta}}{2\omega_1}$
  - If $\Delta = 0$: The equation has exactly one real root:
    * $\omega_4 = \frac{-\omega_2}{2\omega_1}$

---

[1]We consider the full vocabulary minus a set of tokens such as spaces, arithmetic symbols $\{+, -, *, /, =\}$, etc.

– if $\Delta < 0$: The equation has two complex conjugate roots.

* $\omega_4 = \frac{-\omega_2 \pm i\sqrt{\Delta}}{2\omega_1}$

- **Example.** Question. *Solve this quadratic equation:* $x^2 + x - 2 = 0$. Answer. *To solve this quadratic equation, we first calculate the discriminant* $\Delta$: $\Delta = \omega_2^2 - 4\omega_1\omega_3 = 1^2 - 4(1)(-2) = 9$. *Since* $\Delta > 0$, *the equation has two distinct real roots, calculated using the quadratic formula:* $x = \frac{-\omega_2 \pm \sqrt{\Delta}}{2a}$. *Substituting the values, we get: root 1:* $\frac{-1+\sqrt{9}}{2} = 1$, *root 2:* $\frac{-1-\sqrt{9}}{2} = -2$.

**First-order single equation.** We consider a single equation that we solve for a certain variable. To ensure simplicity, we focus on first-order equations, which can be solved through straightforward algebraic transformations without the need for advanced techniques. We generate the answer using SymPy. The equation $\omega_A$ consists of terms on both sides, $\omega_A := \sum_{i=1}^n a_i \cdot \omega_i = \sum_{i=1}^n b_i \cdot \omega_i$, for a variable $\omega_k$.

- Question. *Solve for the variable* $\omega_k$ *in the equation:* $\omega_A$.
- Answer. $\omega_k = \frac{\sum_{i \neq k} \omega_i \cdot (b_i - a_i)}{a_k - b_k}$.
- **Example.** Question. *Solve for the variable* $x$ *in* $13 \cdot x \cdot y + 24 \cdot y^2 = 17 \cdot x \cdot z + 12 \cdot y \cdot z$. Answer. $x = \frac{12 \cdot y \cdot (z - 2 \cdot y)}{13 \cdot y - 17 \cdot z}$.

**Simplify expression.** We want the model to reduce an expression $\omega_A := \sum_{i=1}^n a_i \cdot \omega_i + \sum_{i=1}^n b_i \cdot \omega_i$ to have it in its canonical form, by grouping terms and simplifying some others $\omega_B := \sum_{i=1}^n (a_i + b_i) \cdot \omega_i$. We generate the answer using SymPy.

- Question. *Simplify the following expression:* $\omega_A$.
- Answer. *By grouping terms and eliminating common factors, we get the simplified expression:* $\omega_B$.
- **Example.** Question. *Simplify the following expression:* $-9 \cdot x^2 - 8 \cdot y + 5 \cdot y$. Answer. *By grouping terms:* $-9 \cdot x^2 - 3 \cdot y$.

**Resistors equations.** We consider specific equations that appear in the context of electrical circuits, where resistors are connected in parallel or in series. In the first case, we isolate a variable in an equation of the form $\omega_A := \omega_1 = \omega_2 \times \sum_{i=3}^n \omega_i$:

- Question. *Solve for the variable* $\omega_k$ *in* $\omega_A$.
- Answer. $\omega_k = \frac{\omega_1}{\omega_2} - \sum_{3 \leq i \leq n; i \neq k} \omega_i$
- **Example.** Question. *Solve for the variable* $x$ *in* $u = v \times (w + x)$. Answer. $x = \frac{u}{v} - w$.

In the second case, we isolate a variable in an equation of the form $\omega_B := \omega_1 = \omega_2/(\sum_{i=3}^n 1/\omega_i)$:

- Question. *Solve for the variable* $\omega_k$ *in* $\omega_B$.
- Answer. $\omega_k = 1/\left(\frac{\omega_2}{\omega_1} - \sum_{3 \leq i \leq n; i \neq k} 1/\omega_i\right)$
- **Example.** Question. *Solve for the variable* $x$ *in* $u = v/(1/w + 1/x)$. Answer. $x = 1/(v/u - 1/w)$.

## B.2   Rules Used in Section 4

**Distributivity.** We expand an expression $\omega_A$ consisting of the product of two brackets $\omega_A = (\sum_{i=1}^n \omega_i) * (\sum_{j=1}^m \omega_{n+j})$, to obtain the distributed expression $\omega_B = \sum_{i=1}^n \sum_{j=1}^m \omega_i * \omega_{n+j}$.

- Question. *Expand this expression:* $\omega_A$.
- Answer. *By the distributivity property:* $\omega_B$.
- **Example.** Question. *Expand this expression:* $(-5 + soccer - dog) * (blue - sky)$. Answer. *By the distributivity property:* $-5 * blue + 5 * sky + soccer * blue - soccer * sky - dog * blue + dog * sky$.

**Single equation manipulation.** We consider two rules. The first consists in computing an affine transformation $\omega_3 * \omega_A + \omega_4$ of an equation $\omega_A$ of the form $\omega_1 = \omega_2$, namely $\omega_3 * \omega_1 + \omega_4 = \omega_3 * \omega_2 + \omega_4$. The second rule consists in "simplifying" an equation $\omega_A$ of the form $\omega_1 = \omega_2$ and performs three steps at once: putting the equation in standard form [2], canceling out terms on both sides, and factorizing the remaining terms. We use the convention that equations are always encapsulated within two semicolons, in order to help the model recognize equations.

- Questions. a) *Assumptions: E1 ;$\omega_A$;. Compute: $\omega_3 * E1 + \omega_4$.* b) *Simplify: ;$\omega_A$;.*
- Answers. a-b) *We get: ;$\omega_B$;.*
- Example a). Question. *Assumptions: E1 ;$cat + dog - 2 = tree$;. Compute: $sky * E1 + 8$.* Answer. *We get: ;$sky * cat + sky * dog - sky * 2 + 8 = sky * tree + 8$;.*
- Example b). Question. *Simplify the equation: ;$7 * dog + sky * cat + sky * dog - sky * 2 + 8 = sky * tree + 7 * dog + 8 - blueberry$;.* Answer. *We get: ;$sky * (cat + dog - 2 - tree) + blueberry = 0$;.*

**Commutativity.** We study the commutativity of an operation $\odot_A \in \{*, \pm\}$ in an expression $\omega_A := \omega_1 \odot_1 \omega_2 \odot_2 \cdots \odot_{n-1} \omega_n$. If $\odot_A = *$, then $\odot_i = * \; \forall i$, and if $\odot_A = \pm$, then each $\odot_i$ is either $+$ or $-$ with equal probability. We specify the two types $\omega_i, \omega_j$ among $n$ to which we want to apply commutativity, and the commuted expression $\odot_B$ is the same as $\odot_A$ but where we switch the positions of $\omega_i, \omega_j$. If one of the latter appears more than once in the expression (i.e. $\omega_i = \omega_k$ or $\omega_j = \omega_k$ for some $k \neq i, j$), we choose the first from the left.

- Question. *Apply the commutativity property of $\odot_1$ to $\omega_1, \omega_2$ in $\omega_A$.*
- Answer. *By the commutativity property of $\odot_1$: $\omega_A = \omega_B$.*
- Example. Question. *Apply the commutativity property of $*$ to $cat, 5$ in $5 * cat * soccer$.* Answer. *We get: $cat * 5 * soccer$.*

**Division.** We consider three properties of the division: $\frac{\omega_1 \odot_1 \omega_2}{\omega_3} = \frac{\omega_1}{\omega_3} \odot_1 \frac{\omega_2}{\omega_3}$, where $\odot_1 \in \{+, -\}$; $\frac{\omega_1 * \omega_2}{\omega_3 * \omega_4} = \frac{\omega_1}{\omega_3} * \frac{\omega_2}{\omega_4}$; $\frac{\frac{\omega_1}{\omega_2}}{\omega_3} = \frac{\omega_1}{\omega_2 * \omega_3}$.

- Question. *Use a fundamental property of the division in $\omega_A$.*
- Answer. *By a property of the division: $\omega_B$.*
- Example. Question. *Use a fundamental property of the division in $(-sky * blue)/(tree * -cloud)$.* Answer. *By a property of the division: $(-sky)/(tree) * (blue)/(-cloud)$.*

**Exponentiation.** We consider five properties of the exponentiation: $\omega^0 = 1$; the definition of exponentiation $\omega^n = \underbrace{\omega * \cdots * \omega}_{n \text{ times}}$ if $n$ is a positive integer, the reciprocal of the latter if $n$ is a negative integer; if $n, m$ are signed integers: $\omega^n * \omega^m = \omega^{n+m}$; $\omega_1^n * \omega_2^n = (\omega_1 * \omega_2)^n$; $\frac{\omega^n}{\omega^m} = \omega^{n-m}$.

- Questions. a) *Apply the definition of the exponentiation to: $\omega_A$.* b) *Give the value of: $\omega^0$.* c) *Use a fundamental property of the exponentiation: $\omega_A$.*
- Answers. a) *By definition of the exponentiation: $\omega_B$.* b-c) *By a property of the exponentiation: $\omega_B$..*
- Examples.

  Question. *Use a fundamental property of the exponentiation: $(bluesky^{-3})/(bluesky^9)$.* Answer. *By a property of the exponentiation: $bluesky^{-12}$.*

  Question. *Apply the definition of the exponentiation to: $bluesky^3$.* Answer. *By definition of the exponentiation: $bluesky * bluesky * bluesky$.*

  Question. *Give the value of: $bluesky^0$.* Answer. *By fundamental property of the exponentiation: $bluesky^0 = 1$.*

---

[2]that is, in the form $\omega_1 - \omega_2 = 0$.

**Variable evaluation.** We teach the model to substitute $k \geq 0$ types in a given equation, based on some assumptions on these types (if $k = 0$, nothing occurs). Precisely, assume that we have an equation $\omega_A$ of the form $\omega_1 \odot_1 \omega_2 \odot_2 \cdots \odot_{n-1} \omega_n = \omega_{n+1} \odot_{n+1} \omega_{n+2} \odot_{n+2} \cdots \odot_{n+m-1} \omega_{n+m}$, and that by assumption $\omega_{n_i} = \omega'_{n_i}$ for a set of indexes $\{n_i\}$ and some types $\omega'_{n_i}$. Then, the new equation $\omega_B$ is the same as $\omega_A$ but with $\omega'_{n_i}$ replaced by $\omega_{n_i}$.

- Question. *Assumptions: $\omega_{n_1} = \omega'_{n_1}, \ldots, \omega_{n_k} = \omega'_{n_k}$. Based on the assumptions, evaluate $\omega_A$.*

- Answer. *The evaluated expression: $\omega_B$.*

- Example. Question. *Assumptions: $sky = 2, blueberry = cat$. Based on the assumptions, evaluate $dog - tree + sky = blueberry$.* Answer. *The evaluated expression: $dog - tree + 2 = cat$.*

**Remarkable identities.** We consider the remarkable identity $(\omega_1 \odot_1 \omega_2)^2 = \omega_1^2 + \omega_2^2 \odot_1 2\omega_1\omega_2$, where $\odot_1 \in \{+, -\}$.

- Question. *Expand this expression: $\omega_A$.*

- Answer. *By the remarkable identity properties: $\omega_B$.*

- Example. Question *Expand this expression: $(sky - blueberry)^2$.* Answer. *By the remarkable identity properties: $sky^2 + blueberry^2 - 2 * sky * blueberry$.*

**Combination of two equations.** We consider two skills. The first skill consists in computing an affine transformation $\omega_5 * \omega_A + \omega_6 * \omega_B$ of two equations $\omega_A, \omega_B$ of respective forms $\omega_1 = \omega_2$ and $\omega_3 = \omega_4$. Here, $\omega_1, \omega_2, \omega_3, \omega_4$ are expression types. This yields the equation $\omega_5 * \omega_1 + \omega_6 * \omega_3 = \omega_5 * \omega_2 + \omega_6 * \omega_4$. The second skill consists in being able to determine whether two equations are equivalent. This is done in several steps: first, the two equations are put into standard form. Then, we compute the difference of their left-hand sides. If we get $0 = 0$, then we conclude that the two equations are equivalent, otherwise they are not as the left-hand residual is not zero. In the latter case, the residual is provided in factorized form.

**Systems of equations.** Remember that Gaussian elimination is a method to solve a system of $n$ linear equations in $k$ steps. Each step transforms the system at step $i$ to a new, simpler system at step $i + 1$. Our aim is to teach the model to perform any step $i \to i + 1$. For this, we create a system of $k$ equations with $n \geq k$ variables. The latter are always taken to be of the $m$-token base type. Then, we generate a matrix $M$ of coefficients of size $k \times (n + 1)$ associated with the system, where the coefficients are arbitrary types:

$$M = \begin{bmatrix} a_{11} & a_{12} & \ldots & a_{1n} & | & b_1 \\ a_{21} & a_{22} & \ldots & a_{2n} & | & b_2 \\ \vdots & \vdots & \ddots & \vdots & | & \vdots \\ a_{k1} & a_{k2} & \ldots & a_{kn} & | & b_k \end{bmatrix}$$

Let $x_1, \ldots, x_n$ be the variables, thus our system of equations at step 0 is:

$$a_{11}x_1 + a_{12}x_2 + \ldots + a_{1n}x_n = b_1$$
$$a_{21}x_1 + a_{22}x_2 + \ldots + a_{2n}x_n = b_2$$
$$\vdots$$
$$a_{k1}x_1 + a_{k2}x_2 + \ldots + a_{kn}x_n = b_k$$

We then apply a step of Gaussian elimination, which is a method that transforms the coefficient matrix of the system into row-echelon form and then back-substitutes. First, we perform row operations to transform the augmented matrix into row-echelon form. The goal is to create zeros below the diagonal elements. To do this, we first divide the first row $L_1$ by the coefficient of $x_1$ and then $\forall i \in [2, k]$ we update the expression of the $i^{th}$ row $L_i$, $L_i \to L_i - a_i \cdot L_1$. This eliminates the $x_1$ variable from equation $i$. Then we do the same $\forall i \in [2, k], \forall j \in [i, k]$ we update $L_j \to L_j - a_j \cdot L_i$. This eliminates variable $i$ from each equation $j$. At the end of this first phase, we obtain a new matrix

of coefficients:

$$M' = \begin{bmatrix} 1 & a'_{12} & \dots & a'_{1n} & \vline & b'_1 \\ 0 & 1 & \dots & a'_{2n} & \vline & b'_2 \\ \vdots & \vdots & \ddots & \vdots & \vline & \vdots \\ 0 & 0 & \dots & 1 & \vline & b'_k \end{bmatrix}$$

Back substitution is then performed to express variables in terms of other variables and coefficients such that the new system is simpler.

We store in our data all steps $i \to i+1$. This gives us a set of pairs $(step_i, step_{i+1})$ with $step_i$ the state of the system at step $i$ of Gaussian elimination. From each such pair, we construct two kinds of textual data:

- **One step Gaussian elimination** The question contains $step_i$ and the answer contains $step_{i+1}$.

- **One "clean" step Gaussian elimination** Because our variables and coefficients in the system are essentially $m$-tokens, working performing Gaussian elimination can yield very large equations after a few steps. Thus, later steps will by construction always contain coefficients that are complex mathematical equations. In order to remediate to this issue, we replace the coefficients appearing in front of the variables in step $i$ by simpler coefficients, namely $m$-tokens. Concretely, we will replace large coefficients of the form $\frac{dog - cat*2}{tree*house}$ by a single $m$-token. We call $step_i^{clean}$ the simplification of $step_i$. The question contains $step_i^{clean}$ and the answer contains the corresponding $step_{i+1}^{clean}$.

Systems are built according to this format: $;[; equation_1; ; equation_2; \dots ; equation_n; ];$. The idea behind this convention is to help the model spot equations using the symbol $;$. Below we provide an example of Gaussian elimination:

- Question. *The variables: $dog, sky$. Perform one step of Gaussian elimination*: $;[;-9 * dog - cat * sky = -blueberry;;3 * dog - tree * sky = -6;];$.

- Answer. *We get:* $;[;dog + (cat/9) * sky = (blueberry/9);;3 * dog - tree * sky = -6;];$.

- Example. Question. *The variables: $dog, sky$. Perform one step of Gaussian elimination:* $;[;-9 * dog - cat * sky = -blueberry;;3 * dog - tree * sky = -6;];$. Answer. *We get:* $;[;dog + (cat/9) * sky = (blueberry/9);;3 * dog - tree * sky = -6;];$.

# C Bottom-Up Generalization: Going from Mathematical Rules to Word Problems

Below are the hyperparameters used during our experiments:

- One epoch of training.
- Quantized fine-tuning with low-rank adaption (QLoRA Dettmers et al. (2024)) with a rank of 256 and a dropout of 0.1, and a quantization of 4 bits for the weights.
- Learning rate of $10^{-5}$ with cosine decay.
- AdamW optimizer quantized with 8 bits, weight decay of 0.001 and gradient clipping of 0.3.
- Batch size of 16.

We run our experiments on 4 Nvidia A10 GPUs with 24GB of video memory on an AWS *g5.12xl* instance, thus fitting one replica of the model on each GPU.

## C.1 Quadratic Polynomials

We generate 100 examples of quadratic polynomial problems by systematically varying the surface type (e.g., swimming pools, fields), the names of the owners (e.g., Jim, Alice), the number of each surface (square, rectangle, triangle, parallelogram) as well as the known length (for rectangle, triangle, parallelogram). We ensure that the discriminant of all the polynomials are perfect squares. This constraint avoids evaluating the model's ability to compute square roots.

We provide an example of prompt and response for our problems (Figure 6).

Figure 6: Example of a word problem and response - quadratic polynomials.

> **Prompt**: Jim has a total of 2 swimming pools. The first swimming pool is a square, with an unknown side length x. The second is a rectangle with one side measuring 1 meters and the other being the unknown side length x. The total area covered by these swimming pools is 2 square meters. What is the unknown side length x?
>
> **Response**: The total area is the sum of the area of the square and the rectangle. The area of the square is $x^2$, the area of the rectangle is $x$, and the total area of all fields is 2 square meters. Therefore, the equation to solve is: $x^2 + x = 2$. To solve the quadratic equation, we first calculate the discriminant $\Delta$: $\Delta = b^2 - 4ac = 1^2 - 4(1)(-2) = 9$. Since $\Delta > 0$, the equation has two distinct real roots, calculated using the quadratic formula: $x = \frac{-b \pm \sqrt{\Delta}}{2a}$. Substituting the values, we get: root 1: $\frac{-1+\sqrt{9}}{2} = 1$, root 2: $\frac{-1-\sqrt{9}}{2} = -2$. Only one root is positive, so the solution is $x = 1$.

In our experiments, the baseline model demonstrated suboptimal performance. To address this, we introduced a 3-shot prompting that explicitly detailed the steps for calculating the roots, leading to a marginal improvement in performance, though it remained relatively low ($\approx 10\%$). While the baseline successfully formulates the quadratic polynomial equation, it fails to consistently apply the correct approach for solving it. Instead of computing the discriminant, the baseline frequently attempts an erroneous factorization, which reduces its overall effectiveness.

We fine-tune the baseline model in order to improve its ability to find the roots of the polynomial by computing the discriminant. For training, we generate a dataset based on three mathematical rules: **quadratic polynomials** (200,000 examples), **simplify expression** (100,000 examples), and **single equation manipulation** (100,000 examples). These rules are detailed in section B.1, B.2. Additionally, we incorporate 250,000 examples from the Orca dataset Mukherjee et al. (2023) as a regularizer during training. A larger dataset is generated for quadratic polynomials due to the greater complexity of the task.

In the training data, we ensure that the discriminant of the polynomials are always perfect squares, preventing the model from learning how to compute square roots.

Let's consider the quadratic polynomial expression $\omega_1 \times \omega_4^2 + \omega_2 \times \omega_4 + \omega_3$. Initially, the training data included all possible integer values for $\omega_1$ and $\omega_2$ between $-10$ and 10, and for $\omega_3$ between

−100 and 100. However, this range proved too broad, making it difficult for the model to learn effectively. As a result, the fine-tuned model frequently miscalculated the discriminant. To address this issue, we narrowed the parameter distribution, restricting $\omega_1$ and $\omega_2$ to positive integers, and $\omega_3$ to negative values, better aligning the training data with our data at test time.

## C.2 Physics Problems Involving Resistors

We fine-tuned the model on 100,000 examples with equations akin to those necessary to solve resistor circuits. 50,000 examples correspond to resistors in series, and 50,000 examples correspond to resistors in parallel. We provide an example of prompt and response for a resistor circuit in parallel (Figure 7). We also add 25,000 examples from the Orca dataset Mukherjee et al. (2023) as a regularizer during training.

For this particular problem we only consider symbolic solutions, thus we do not introduce numerical values for the variables. We evaluate the model's response by extracting the symbolic solution and comparing it to the correct one.

Figure 7: Example of a prompt and response for a resistor circuit.

**Prompt**: You have a circuit with the following resistors: [R1 ∥ R2]. Given that the current flowing through the circuit is I amp and the voltage across the circuit is U volts, express the resistance of R1 in terms of the other variables.
**Response**: Using properties of resistors in series and parallel, and Ohm's law, we can determine that the equation governing the circuit is U = I/(1/R2 + 1/R1). Isolating the unknown, we get the symbolic solution R1 = 1/(I/U - 1/R2).

The error modes are similar for the baseline and fine-tuned models. The model is always able to find the correct equation, but the fine-tuned model makes less mistakes. These mistakes are mainly due to sign errors or failing to inverse a fraction (i.e. if $1/x = A$ then $x = 1/A$).

## C.3 Fruit Baskets

For the fruit baskets examples we trained a model mainly on two mathematical rules: **First-order single equation** (150,000 examples), and **Simplify expression** (150,000 examples).We selected the hyperparameters for these rules (such as the number of terms and variables) to ensure that the generated data aligns with the format of equations typically found in the fruit baskets problem. We refer to section B.2 for the full details about these rules. Additionally, we incorporate 150,000 examples from the Orca dataset Mukherjee et al. (2023) as a regularizer during training. The Fruit Baskets problem requires setting up an equation, simplifying it to its canonical form, and solving for a specific variable. By fine-tuning the model on these rules, it can generalize and effectively simplify and solve the equation in Fruit Baskets problems.

**"Numerical" Fruit Baskets.** We also considered assigning numerical values to the variables in that problem, requiring the model to perform calculations to find a numerical solution instead of a symbolic one. We illustrate this in figure 9. We found that the models' performances were similar in the symbolic and numerical cases. This is showcased in table 3.

Table 3: Accuracy (%) - fruit basket problems (3-shot).

| Model | Symbolic | | Numerical | |
|---|---|---|---|---|
| | 2 Fruits | 3 Fruits | 2 Fruits | 3 Fruits |
| **Llama-3 8B Instruct** | 19 | 0 | 21 | 6 |
| **Llama-3 8B Fine-tuned** | 35 | 13 | 36 | 13 |

Figure 8: Example of a prompt and response for a symbolic fruit basket problem.

**Prompt**: Alice and Bob went to the grocery store and bought the following items:
- bananas: Alice bought $q_{A_1}$, and Bob bought $q_{B_1}$. The price of a single one is $p_1$.
- blueberries: Alice bought $q_{A_2}$ where $q_{A_2} = 2 \times p_1$, and Bob bought $q_{B_2}$ where $q_{B_2} = 8 \times q_{A_1}$. The price of a single one is $p_2$ where $p_2 = 5 \times q_{B_1}$.
Both ended up paying the same total price. Find the price of bananas in terms of $q_{A_1}$ and $q_{B_1}$.
**Response**: Since both Alice and Bob ended up paying the same total price for the fruits they bought, we can set up the following equation:

$$q_{A_1} \times p_1 + q_{A_2} \times p_2 = q_{B_1} \times p_1 + q_{B_2} \times p_2$$

We substitute the known relationships:

$$q_{A_1} \times p_1 + 2 \times p_1 \times 5 \times q_{B_1} = q_{B_1} \times p_1 + 8 \times q_{A_1} \times 5 \times q_{B_1}$$

Let $x = p_1$, $y = q_{A_1}$, and $z = q_{B_1}$. The equation becomes:

$$y \times x + 2 \times x \times 5 \times z = z \times x + 8 \times y \times 5 \times z$$

Simplifying:

$$y \times x + 10 \times x \times z = z \times x + 40 \times y \times z$$

Solving for $x$, we get:

$$x = \frac{40 \times y \times z}{y + 9 \times z}$$

Since $x = p_1$, $y = q_{A_1}$, and $z = q_{B_1}$, we have:

$$p_1 = \frac{40 \times q_{A_1} \times q_{B_1}}{q_{A_1} + 9 \times q_{B_1}}$$

Figure 9: Example of a prompt and response for a numerical fruit basket problem.

**Prompt**: Alice and Bob went to the grocery store and bought the following items:
- bananas: Alice bought $q_{A_1}$, and Bob bought $q_{B_1}$. The price of a single one is $p_1$.
- blueberries: Alice bought $q_{A_2}$ where $q_{A_2} = 2 \times p_1$, and Bob bought $q_{B_2}$ where $q_{B_2} = 8 \times q_{A_1}$. The price of a single one is $p_2$ where $p_2 = 5 \times q_{B_1}$.
Both ended up paying the same total price. Find the price of bananas, given that $q_{A_1} = 2$ and $q_{B_1} = 1$.
**Response**: Since both Alice and Bob ended up paying the same total price for the fruits they bought, we can set up the following equation:

$$q_{A_1} \times p_1 + q_{A_2} \times p_2 = q_{B_1} \times p_1 + q_{B_2} \times p_2$$

We substitute the known relationships:

$$q_{A_1} \times p_1 + 2 \times p_1 \times 5 \times q_{B_1} = q_{B_1} \times p_1 + 8 \times q_{A_1} \times 5 \times q_{B_1}$$

Given that $q_{A_1} = 2$ and $q_{B_1} = 1$, the equation becomes:

$$1 \times p_1 + 2 \times p_1 \times 5 \times 2 = 2 \times p_1 + 8 \times 1 \times 5 \times 2$$

Simplifying:

$$p_1 + 20 \times p_1 = 2 \times p_1 + 80$$

Solving for $p_1$, we get:

$$p_1 = \frac{80}{19}$$

## C.4 Benchmark Performance

Table 4: Benchmark performance.

| Model | GSM8K (5-shot) | MMLU (0-shot) | ARC-Challenge (0-shot) | HellaSwag (0-shot) |
|---|---|---|---|---|
| Llama-3 8B Instruct | 64.1 | 58.6 | 49.6 | 62.7 |
| Llama-3 8B Fine-tuned (Resistors) | 65.4 | 60.8 | 51.8 | 65.1 |
| Llama-3 8B Fine-tuned (Polynomials) | 68.9 | 60.8 | 51.4 | 66.4 |
| Llama-3 8B Fine-tuned (Fruits) | 69.1 | 60.9 | 51.7 | 65.7 |

The performance on these benchmarks remains relatively stable after fine-tuning, demonstrating that the newly acquired skills did not disrupt the model's prior knowledge.

# D  Top-Down Generalization: Increasing the Mathematical Complexity of the Task

We fine-tune Llama-2 7B Chat Touvron et al. (2023) on *all* the data described in section B.2 (and that data only) and evaluate the model's ability to perform the following rules: *distributivity, commutativity, division, exponentiation, variable evaluation, remarkable identities, single equation and two equations manipulation*. The *research questions* that we want to answer are:

- Can the model learn the rules that it has been trained on?
- Can the model generalize the rules that it has been trained on?
- Can the model use the rules when prompted differently than during training?
- Can the model use combinations of the individual rules that it has been trained on?

Hyperparameters and compute used during experiments are detailed in section D.6. We will essentially consider two versions of our fine-tuned model: one trained on distributivity data only (section D.1), and one trained on all mathematical rules (sections D.2, D.3, D.5). In section D.4, we evaluate our trained models on general knowledge benchmarks in order to check that performance remains relatively stable.

## D.1  Distributivity Data: Impact of Training on Various Tokenizer Vocabulary Sizes on Generalization Performance

We generate training data of 2 million examples of the distributivity rule as described in section B.2, where each bracket contains at most 5 types $\omega_i$: the number of variables is a random integer from 1 to 5. Each type is either an integer (probability 10%) or a $k$-token (probability 90%), where $k \in [1, 3]$. The variables appearing in the equations take value in subsets of increasing tokenizer vocabulary sizes. For this, we split the tokenizer vocabulary of 32,000 tokens into partitions of increasing sizes (1%, 10%, 50%, 75%, 95% and 100%). We evaluate the model both on the complement of its training vocabulary (i.e. on examples that it has never seen during training), as well as on its training vocabulary. For each example, we report a score of 1 if the model prediction matches exactly the expected answer (i.e., the two strings are equal), and 0 otherwise. We present experimental results in figures 4, 10.

We find that training on larger vocabulary sizes improves the ability of the model to generalize distributivity to unseen variable names as well as to increasing the number of variables. The model's performance on the training vocabulary is stable accross different vocabulary sizes, and significantly higher than the performance on the complement vocabulary, even for the largest vocabulary size. In terms of generalization, the models reach decent performance on their training vocabulary (consider that distributing 7 times 7 terms leads to 49 terms). In particular, we observe that the model struggles with some particular tokens (e.g. chinese or cyrillic characters). Other than that, it makes mistakes on very large expressions, forgetting chunks of the expression towards the end, or confusing signs.

In figure 11 we evaluate the model trained on the full vocabulary on a subset of the vocabulary restricted to latin alphabet (lowercase and uppercase). The model's performance increases in the generalization domain, suggesting some sensibility to certain tokens.

Figure 10: Validation accuracy on the distributivity rule for different vocabulary sizes. Each model is evaluated on its training vocabulary. From left to right, from top to bottom: $x = 1, 10, 50, 75, 95,$ 100. The dashed lines delimit the parameters seen during training from those unseen.

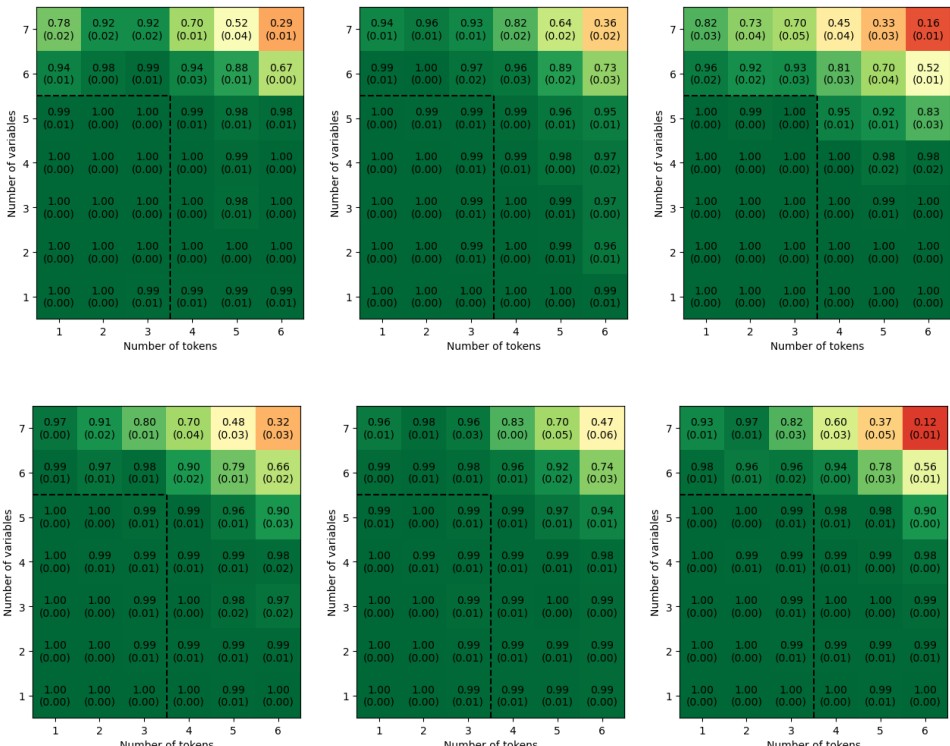

Figure 11: Validation accuracy on the distributivity rule for the full vocabulary with only latin characters. The dashed lines delimit the parameters seen during training from those unseen.

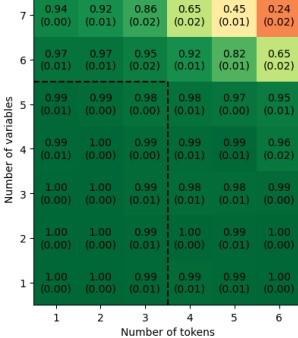

### D.2 Solving Systems of Equations by Recursive Call

System solving is a difficult task for most language models, as it requires the elaboration of a series of reasoning steps. Language models have a finite context window, which constrains the amount of information they can process. Furthermore, numerous variables must be maintained and manipulated over multiple steps. Language models struggle with long-range dependencies, where the relevant information from earlier steps needs to be accurately memorized and applied in later steps. Even state-of-the-art models have difficulty solving systems with more than two variables and equations. To overcome these problems, our idea is to break down system solving into elementary steps and teach our model to correctly transition from each one of the steps to the next (as opposed to training it on the full resolution). The training data contains examples where the system is already resolved (termination condition), and when it is the case, the model recognizes that there is nothing left to do and returns *The system is already simplified*.

The detailed construction of the corresponding data from steps $i \rightarrow i + 1$ of the Gaussian elimination is provided in section B.2. We considered systems from 1 to 5 variables. For each step transition $i \rightarrow i + 1$, we report a score of 1 if the model predicted string matches exactly the ground truth, and 0 otherwise. We obtain an average score of **88.5** $\pm 0.5$ **%**. The latter was computed over 100 examples and 2 random seeds. We provide a detailed example of system resolution in figure 5.

### D.3 Experiments on All Mathematical Rules

We fine-tune our model on all the mathematical rules detailed in section B. To evaluate the models, we considered eight different configurations:

- **4 train configurations**: We use the same hyperparameters as the training dataset: 3 to 5 variables per instance. We consider two vocabulary settings:
  - **Full vocabulary**: The full Llama-2 vocabulary. Each term is a concatenation of 1 to 3 Llama-2 tokens.
    * With integers: 10% of the variables in an expression are integers, while the rest are a concatenation of Llama-2 tokens.
    * Without integers: All the variables are a concatenation of Llama-2 tokens.
  - **Restricted vocabulary**: The vocabulary consists of Latin and Greek letters, one token per variable, and no integers allowed.
    * With digits: The vocabulary consists of Latin and Greek letters and a concatenation of these letters with numbers from 0 to 9 (e.g., $\alpha_0$).
    * Without digits: The vocabulary consists of only Latin and Greek letters without any numbers.
- **4 test configurations**: We increase the number of variables to 6 to 7 variables per instance. We consider two vocabulary settings:
  - **Full vocabulary**: The full Llama-2 vocabulary. Each term is a concatenation of 3 to 5 Llama-2 tokens.
    * With integers: 10% of the variables in an expression are integers, while the rest are a concatenation of Llama-2 tokens.
    * Without integers: All the variables are a concatenation of Llama-2 tokens.
  - **Restricted vocabulary**: Same as the training configuration

The goal of excluding integers in some configurations is to ensure robust evaluation of symbolic reasoning, avoiding cases where models might output correct numerical results without demonstrating an understanding of the underlying properties. For example, in evaluating the division property $(1 + 5)/3 = 1/3 + 5/3$, we expect models to show understanding of this property rather than simply computing numerical expressions. On the other hand, the restricted vocabulary relying on the Latin and Greek alphabetical letters ensures a fair comparison to baseline models likely unfamiliar with unusual tokens in Llama 2's tokenizer. The baseline models considered are the instruct versions of Llemma Azerbayeva et al. (2024), Llama-2, Llama-3, Mistral, and WizardLM Luo et al. (2023). Each model generated responses with a `max_new_tokens` set to 512. This token limit was determined to be sufficient for generating correct answers based on tests with multiple lengths (up to 1500 tokens),

taking into consideration the verbose nature of these models. Each mathematical rule was evaluated on 100 examples over 2 seeds. The results are presented in tables 5, 6, 7 and 8.

Our fine-tuned Llama-2 model consistently outperforms other models across all considered mathematical rules, irrespective of the configuration. Notably, our model demonstrates strong generalization capabilities across various configurations. Baseline models demonstrate moderate performance on the restricted vocabulary and significantly underperform on the full vocabulary, which is expected due to their limited grasp of what a mathematical variable is. In contrast, our model maintains consistently high scores across the configurations. Additionally, we conduct a safety check by verifying string equality as a lower-bound metric to ensure the accuracy of our metrics. This validation process is demonstrated in Table 13.

Given the lengthy nature of the prompts in distributivity and two equations tasks, we decided to add another test configuration with the full vocabulary with integers in which we only increase the number of variables taking it from 6 to 7, while keeping the variables in the same shape as the ones in the training (a concatenation of 1 to 3 tokens). We only test this configuration on the two skills of distributivity and two equations, and the results are presented in table 14.

Table 5: Train configuration. Full vocabulary with integers.

| Model | Distributivity | Commutativity | Division | Exponent-iation | Variable Evaluation | Remarkable Identities | Single Equation | Two Equations |
|---|---|---|---|---|---|---|---|---|
| Llama 2 7B Chat fine-tuned (all) | $97.0 \pm 2.0$ | $99.0 \pm 1.0$ | $99.0 \pm 0.0$ | $99.5 \pm 0.5$ | $96.5 \pm 2.5$ | $98.0 \pm 2.0$ | $98.5 \pm 0.5$ | $99.0 \pm 0.0$ |
| Llama-3 8B Instruct | $0.5 \pm 0.5$ | $9.0 \pm 0.0$ | $2.5 \pm 0.5$ | $52.0 \pm 3.0$ | $31.0 \pm 2.0$ | $24.0 \pm 7.0$ | $0.0 \pm 0.0$ | $0.0 \pm 0.0$ |
| WizardMath 7B | $9.5 \pm 0.5$ | $12.5 \pm 0.5$ | $6.5 \pm 1.5$ | $65.0 \pm 1.0$ | $26.5 \pm 5.5$ | $36.5 \pm 1.5$ | $0.0 \pm 0.0$ | $2.0 \pm 1.0$ |
| Mistral 7B Instruct | $1.5 \pm 0.5$ | $11.4 \pm 3.4$ | $5.0 \pm 1.0$ | $51.0 \pm 5.9$ | $8.0 \pm 4.0$ | $9.0 \pm 3.0$ | $0.5 \pm 0.5$ | $0.0 \pm 0.0$ |
| Llama-2 7B chat | $0.0 \pm 0.0$ | $4.5 \pm 1.5$ | $2.5 \pm 1.5$ | $24.0 \pm 2.0$ | $2.0 \pm 1.0$ | $4.5 \pm 1.0$ | $0.0 \pm 0.0$ | $0.5 \pm 0.5$ |
| llemma 7B | $0.0 \pm 0.0$ | $0.5 \pm 0.5$ | $0.0 \pm 0.0$ | $25.0 \pm 0.0$ | $5.5 \pm 3.4$ | $0.5 \pm 0.5$ | $0.0 \pm 0.0$ | $0.0 \pm 0.0$ |

Table 6: Test configuration. Full vocabulary with integers.

| Model | Distributivity | Commutativity | Division | Exponent-iation | Variable Evaluation | Remarkable Identities | Single Equation | Two Equations |
|---|---|---|---|---|---|---|---|---|
| Llama 2 7B Chat fine-tuned (all) | $24.0 \pm 3.0$ | $97.5 \pm 2.5$ | $98.5 \pm 0.5$ | $100.0 \pm 0.0$ | $98.5 \pm 0.5$ | $99.0 \pm 1.0$ | $94.5 \pm 3.5$ | $73.5 \pm 4.5$ |
| Llama-3 8B Instruct | $0.0 \pm 0.0$ | $1.0 \pm 1.0$ | $0.5 \pm 0.5$ | $32.5 \pm 5.5$ | $8.5 \pm 1.5$ | $14.5 \pm 0.5$ | $0.0 \pm 0.0$ | $20.0 \pm 2.0$ |
| WizardMath 7B | $0.0 \pm 0.0$ | $1.0 \pm 1.0$ | $6.5 \pm 0.5$ | $55.5 \pm 1.5$ | $7.5 \pm 1.5$ | $26.5 \pm 7.5$ | $0.0 \pm 0.0$ | $21.0 \pm 2.0$ |
| Mistral 7B Instruct | $0.0 \pm 0.0$ | $0.0 \pm 0.0$ | $4.0 \pm 0.0$ | $45.0 \pm 2.5$ | $0.5 \pm 0.5$ | $4.5 \pm 1.5$ | $0.0 \pm 0.0$ | $17.0 \pm 1.0$ |
| Llama-2 7B chat | $0.0 \pm 0.0$ | $0.0 \pm 0.0$ | $0.0 \pm 0.0$ | $19.5 \pm 4.5$ | $0.0 \pm 0.0$ | $1.5 \pm 0.5$ | $0.0 \pm 0.0$ | $38.0 \pm 3.0$ |
| llemma 7B | $0.0 \pm 0.0$ | $2.0 \pm 1.0$ | $1.5 \pm 1.5$ | $44.0 \pm 0.0$ | $4.0 \pm 2.0$ | $7.5 \pm 0.5$ | $0.0 \pm 0.0$ | $7.5 \pm 0.5$ |

Table 7: Train configuration. Restricted vocabulary with digits.

| Model | Distributivity | Commutativity | Division | Exponent-iation | Variable Evaluation | Remarkable Identities | Single Equation | Two Equations |
|---|---|---|---|---|---|---|---|---|
| Llama 2 7B Chat fine-tuned (all) | $89.0 \pm 4.0$ | $98.0 \pm 1.0$ | $100.0 \pm 0.0$ | $100.0 \pm 0.0$ | $98.5 \pm 0.5$ | $100.0 \pm 0.0$ | $99.5 \pm 0.5$ | $98.5 \pm 0.5$ |
| Llama-3 8B Instruct | $30.0 \pm 1.9$ | $44.0 \pm 4.9$ | $16.5 \pm 2.4$ | $60.5 \pm 1.5$ | $91.0 \pm 1.0$ | $36.0 \pm 1.9$ | $1.0 \pm 0.0$ | $13.5 \pm 1.5$ |
| WizardMath 7B | $45.5 \pm 3.5$ | $38.5 \pm 1.5$ | $19.5 \pm 5.5$ | $58.5 \pm 2.5$ | $63.5 \pm 4.5$ | $46.0 \pm 3.0$ | $0.0 \pm 0.0$ | $18.5 \pm 4.5$ |
| Mistral 7B Instruct | $26.5 \pm 2.5$ | $36.5 \pm 1.5$ | $22.0 \pm 0.0$ | $42.0 \pm 3.0$ | $56.0 \pm 4.0$ | $15.5 \pm 4.5$ | $1.0 \pm 0.0$ | $13.0 \pm 2.0$ |
| Llama-2 7B chat | $3.0 \pm 0.0$ | $15.0 \pm 0.0$ | $5.5 \pm 0.5$ | $37.5 \pm 2.5$ | $28.0 \pm 1.9$ | $12.0 \pm 6.0$ | $0.0 \pm 0.0$ | $15.5 \pm 1.5$ |
| llemma 7B | $0.0 \pm 0.0$ | $2.5 \pm 0.5$ | $1.5 \pm 1.5$ | $50.0 \pm 5.0$ | $4.0 \pm 2.0$ | $3.0 \pm 0.0$ | $0.0 \pm 0.0$ | $18.5 \pm 0.5$ |

Table 8: Test configuration. Restricted vocabulary with digits.

| Model | Distributivity | Commutativity | Division | Exponent-iation | Variable Evaluation | Remarkable Identities | Single Equation | Two Equations |
|---|---|---|---|---|---|---|---|---|
| Llama 2 7B Chat fine-tuned (all) | $81.5 \pm 0.5$ | $89.0 \pm 2.0$ | $100.0 \pm 0.0$ | $100.0 \pm 0.0$ | $92.0 \pm 2.0$ | $100.0 \pm 0.0$ | $100.0 \pm 0.0$ | $96.5 \pm 1.5$ |
| Llama-3 8B Instruct | $15.0 \pm 4.0$ | $29.5 \pm 3.5$ | $16.5 \pm 2.5$ | $60.5 \pm 1.5$ | $79.0 \pm 4.0$ | $35.5 \pm 2.5$ | $0.0 \pm 0.0$ | $3.0 \pm 1.0$ |
| WizardMath 7B | $4.0 \pm 0.0$ | $27.0 \pm 4.0$ | $19.5 \pm 5.5$ | $58.5 \pm 2.5$ | $52.5 \pm 2.5$ | $46.0 \pm 3.0$ | $0.5 \pm 0.5$ | $2.5 \pm 0.5$ |
| Mistral 7B Instruct | $2.5 \pm 1.5$ | $18.5 \pm 0.5$ | $22.0 \pm 0.0$ | $42.5 \pm 2.5$ | $52.0 \pm 4.0$ | $15.5 \pm 4.5$ | $1.0 \pm 1.0$ | $1.5 \pm 0.5$ |
| Llama-2 7B chat | $0.0 \pm 0.0$ | $4.5 \pm 0.5$ | $5.5 \pm 0.5$ | $37.5 \pm 2.5$ | $15.5 \pm 1.5$ | $12.0 \pm 6.0$ | $0.0 \pm 0.0$ | $1.0 \pm 1.0$ |
| llemma 7B | $0.0 \pm 0.0$ | $1.0 \pm 1.0$ | $1.5 \pm 1.5$ | $36.5 \pm 2.0$ | $3.0 \pm 1.0$ | $3.0 \pm 0.0$ | $0.0 \pm 0.0$ | $5.0 \pm 1.0$ |

Table 9: Train configuration. Restricted vocabulary without digits.

| Model | Distributivity | Commutativity | Division | Exponent-iation | Variable Evaluation | Remarkable Identities | Single Equation | Two Equations |
|---|---|---|---|---|---|---|---|---|
| Llama 2 7B Chat fine-tuned (all) | $96.0 \pm 2.0$ | $99.5 \pm 0.5$ | $100.0 \pm 0.0$ | $100.0 \pm 0.0$ | $95.5 \pm 1.5$ | $100.0 \pm 0.0$ | $98.5 \pm 1.5$ | $97.5 \pm 0.5$ |
| Llama-3 8B Instruct | $36.0 \pm 6.0$ | $43.5 \pm 1.5$ | $21.5 \pm 0.5$ | $79.5 \pm 2.5$ | $88.0 \pm 2.0$ | $45.0 \pm 1.0$ | $2.5 \pm 0.5$ | $24.5 \pm 5.5$ |
| WizardMath 7B | $48.5 \pm 3.5$ | $47.0 \pm 3.0$ | $24.5 \pm 0.5$ | $85.0 \pm 3.0$ | $68.5 \pm 0.5$ | $72.0 \pm 0.0$ | $2.5 \pm 0.5$ | $22.5 \pm 3.5$ |
| Mistral 7B Instruct | $28.0 \pm 3.0$ | $31.5 \pm 6.5$ | $24.5 \pm 2.5$ | $70.5 \pm 0.5$ | $54.0 \pm 5.0$ | $41.0 \pm 7.0$ | $1.0 \pm 1.0$ | $19.5 \pm 2.5$ |
| Llama-2 7B chat | $7.0 \pm 0.0$ | $14.5 \pm 0.5$ | $7.0 \pm 1.0$ | $45.0 \pm 4.0$ | $31.0 \pm 4.0$ | $1.0 \pm 1.0$ | $0.0 \pm 0.0$ | $37.5 \pm 1.5$ |
| llemma 7B | $2.5 \pm 0.5$ | $3.5 \pm 0.5$ | $1.5 \pm 1.5$ | $44.0 \pm 0.0$ | $9.5 \pm 1.5$ | $7.5 \pm 0.5$ | $0.5 \pm 0.5$ | $16.0 \pm 3.0$ |

Table 10: Test configuration. Restricted vocabulary without digits.

| Model | Distributivity | Commutativity | Division | Exponent-iation | Variable Evaluation | Remarkable Identities | Single Equation | Two Equations |
|---|---|---|---|---|---|---|---|---|
| Llama 2 7B Chat fine-tuned (all) | $75.0 \pm 2.0$ | $94.5 \pm 0.5$ | $100.0 \pm 0.0$ | $100.0 \pm 0.0$ | $88.0 \pm 1.0$ | $100.0 \pm 0.0$ | $97.0 \pm 1.0$ | $95.5 \pm 1.5$ |
| Llama-3 8B Instruct | $6.5 \pm 0.5$ | $25.5 \pm 4.5$ | $21.5 \pm 0.5$ | $79.5 \pm 2.5$ | $27.5 \pm 0.5$ | $45.0 \pm 1.0$ | $2.0 \pm 1.0$ | $7.5 \pm 1.5$ |
| WizardMath 7B | $4.0 \pm 1.0$ | $28.0 \pm 2.0$ | $24.5 \pm 0.5$ | $85.0 \pm 3.0$ | $49.0 \pm 1.0$ | $72.0 \pm 0.0$ | $1.0 \pm 0.0$ | $8.0 \pm 1.0$ |
| Mistral 7B Instruct | $7.0 \pm 2.0$ | $19.5 \pm 1.5$ | $24.5 \pm 2.5$ | $71.0 \pm 1.0$ | $42.0 \pm 1.0$ | $41.0 \pm 7.0$ | $1.5 \pm 0.5$ | $4.5 \pm 0.5$ |
| Llama-2 7B chat | $0.0 \pm 0.0$ | $7.0 \pm 0.0$ | $7.0 \pm 1.0$ | $45.0 \pm 4.0$ | $22.0 \pm 0.0$ | $1.0 \pm 1.0$ | $0.0 \pm 0.0$ | $20.0 \pm 2.0$ |
| llemma 7B | $0.0 \pm 0.0$ | $2.0 \pm 1.0$ | $1.5 \pm 1.5$ | $44.0 \pm 0.0$ | $4.0 \pm 2.0$ | $7.5 \pm 0.5$ | $0.0 \pm 0.0$ | $7.5 \pm 0.5$ |

Table 11: Train configuration. Full vocabulary without integers.

| Model | Distributivity | Commutativity | Division | Exponent-iation | Variable Evaluation | Remarkable Identities | Single Equation | Two Equations |
|---|---|---|---|---|---|---|---|---|
| Llama 2 7B Chat fine-tuned (all) | $98.0 \pm 1.0$ | $99.0 \pm 1.0$ | $99.0 \pm 0.0$ | $99.5 \pm 0.5$ | $99.0 \pm 1.0$ | $98.0 \pm 2.0$ | $99.0 \pm 1.0$ | $99.5 \pm 0.5$ |
| Llama-3 8B Instruct | $0.5 \pm 0.5$ | $9.5 \pm 0.5$ | $2.0 \pm 0.0$ | $49.0 \pm 0.0$ | $35.0 \pm 6.0$ | $21.5 \pm 4.5$ | $0.0 \pm 0.0$ | $0.0 \pm 0.0$ |
| WizardMath 7B | $6.5 \pm 0.5$ | $9.5 \pm 0.5$ | $7.0 \pm 2.0$ | $65.0 \pm 0.5$ | $22.5 \pm 4.5$ | $32.5 \pm 0.5$ | $0.0 \pm 0.0$ | $1.0 \pm 0.0$ |
| Mistral 7B Instruct | $3.0 \pm 1.0$ | $8.5 \pm 2.5$ | $5.0 \pm 2.0$ | $51.0 \pm 4.0$ | $5.5 \pm 0.5$ | $8.5 \pm 3.5$ | $0.0 \pm 0.0$ | $0.0 \pm 0.0$ |
| Llama-2 7B chat | $0.0 \pm 0.0$ | $4.5 \pm 0.5$ | $1.5 \pm 1.5$ | $21.0 \pm 1.0$ | $0.5 \pm 0.5$ | $5.5 \pm 0.5$ | $0.0 \pm 0.0$ | $0.0 \pm 0.0$ |
| llemma 7B | $0.0 \pm 0.0$ | $0.5 \pm 0.5$ | $0.0 \pm 0.0$ | $22.5 \pm 1.5$ | $0.0 \pm 0.0$ | $0.5 \pm 0.5$ | $0.0 \pm 0.0$ | $0.0 \pm 0.0$ |

Table 12: Test configuration. Full vocabulary without integers.

| Model | Distributivity | Commutativity | Division | Exponent-iation | Variable Evaluation | Remarkable Identities | Single Equation | Two Equations |
|---|---|---|---|---|---|---|---|---|
| Llama 2 7B Chat fine-tuned (all) | $15.0 \pm 5.0$ | $96.0 \pm 1.0$ | $99.0 \pm 1.0$ | $100.0 \pm 0.0$ | $96.0 \pm 1.0$ | $99.0 \pm 1.0$ | $94.5 \pm 1.5$ | $65.5 \pm 4.5$ |
| Llama-3 8B Instruct | $0.0 \pm 0.0$ | $0.5 \pm 0.5$ | $0.5 \pm 0.5$ | $27.5 \pm 1.5$ | $5.0 \pm 0.0$ | $16.5 \pm 0.5$ | $0.0 \pm 0.0$ | $16.5 \pm 0.5$ |
| WizardMath 7B | $0.0 \pm 0.0$ | $1.0 \pm 1.0$ | $3.0 \pm 2.0$ | $56.0 \pm 1.0$ | $5.5 \pm 3.5$ | $19.5 \pm 6.5$ | $0.0 \pm 0.0$ | $23.5 \pm 2.5$ |
| Mistral 7B Instruct | $0.0 \pm 0.0$ | $0.0 \pm 0.0$ | $1.0 \pm 0.0$ | $42.5 \pm 3.5$ | $0.0 \pm 0.0$ | $3.0 \pm 2.0$ | $0.0 \pm 0.0$ | $15.5 \pm 2.5$ |
| Llama-2 7B chat | $0.0 \pm 0.0$ | $0.0 \pm 0.0$ | $0.5 \pm 0.5$ | $17.0 \pm 2.0$ | $0.0 \pm 0.0$ | $2.5 \pm 1.5$ | $0.0 \pm 0.0$ | $35.5 \pm 2.5$ |
| llemma 7B | $0.0 \pm 0.0$ | $0.0 \pm 0.0$ | $0.0 \pm 0.0$ | $17.0 \pm 2.0$ | $0.0 \pm 0.0$ | $0.0 \pm 0.0$ | $0.0 \pm 0.0$ | $19.5 \pm 2.5$ |

Table 13: Evaluation results on exact predictions. For each example, we report a score of 1 if the model prediction matches exactly the expected answer (i.e., the two strings are equal), and 0 otherwise.

| Configuration | Distributivity | Commutativity | Division | Exponent-iation | Variable Evaluation | Remarkable Identities | Single Equation | Two Equations |
|---|---|---|---|---|---|---|---|---|
| Test, full vocabulary without integers | $15.0 \pm 5.0$ | $96.0 \pm 1.0$ | $99.0 \pm 1.0$ | $100.0 \pm 0.0$ | $96.0 \pm 1.0$ | $99.0 \pm 1.0$ | $94.5 \pm 1.5$ | $29.0 \pm 1.0$ |
| Test, full vocabulary with integers | $24.0 \pm 3.0$ | $97.5 \pm 1.5$ | $98.5 \pm 0.5$ | $100.0 \pm 0.0$ | $98.5 \pm 0.5$ | $99.0 \pm 1.0$ | $94.0 \pm 4.0$ | $31.0 \pm 3.0$ |
| Train, full vocabulary without integers | $99.0 \pm 1.0$ | $99.0 \pm 1.0$ | $99.0 \pm 0.0$ | $99.5 \pm 0.5$ | $99.0 \pm 1.0$ | $98.0 \pm 2.0$ | $99.0 \pm 1.0$ | $97.0 \pm 0.0$ |
| Train, full vocabulary with integers | $98.5 \pm 1.5$ | $99.0 \pm 1.0$ | $99.0 \pm 0.0$ | $99.5 \pm 0.5$ | $98.5 \pm 0.5$ | $98.0 \pm 2.0$ | $98.5 \pm 0.5$ | $97.0 \pm 1.0$ |

Table 14: Test configuration for distributivity and two equations. Full vocabulary with integers

| Model | Test configuration for distributivity and two equations. Full vocabulary with integers | |
|---|---|---|
| | **Distributivity** | **Two Equations** |
| Llama 2 7B Chat fine-tuned (all) | $70.0 \pm 7.0$ | $78.0 \pm 1.0$ |
| Llama-3 8B Instruct | $0.0 \pm 0.0$ | $24.5 \pm 1.5$ |
| WizardMath 7B | $0.0 \pm 0.0$ | $41.0 \pm 2.0$ |
| Mistral 7B Instruct | $0.0 \pm 0.0$ | $19.5 \pm 1.5$ |
| Llama-2 7B chat | $0.0 \pm 0.0$ | $27.0 \pm 4.0$ |
| llemma 7B | $0.0 \pm 0.0$ | $30.5 \pm 0.5$ |

## D.4 Benchmark Performance

Catastrophic forgetting is a well-known issue when fine-tuning large language models, where the model's performance on the pre-training task decreases after fine-tuning on a new task. We evaluate our models on 3 established benchmarks before and after fine-tuning. The results are presented in table 15. While there is a small performance drop after fine-tuning, the model still performs well on the general knowledge tasks.

Table 15: Evaluation on general knowledge tasks. We evaluate on the MMLU Hendrycks et al. (2020), ARC Clark et al. (2018) and HellaSwag Zellers et al. (2019) datasets for general knowledge and language understanding. The evaluation is performed under the framework of Gao et al. (2023).

| Model | MMLU | ARC | HellaSwag |
|---|---|---|---|
| Llama 2 7B Chat pretrained | 46.4 | 64.1 | 57.8 |
| Llama 2 7B Chat fine-tuned (all operations) | 39.7 | 57.6 | 53.7 |
| Llama 2 7B Chat fine-tuned (distributivity) | 37.6 | 57.9 | 54.6 |

## D.5 Some Word Problems

Here we discuss cases where our trained model successfully applies the mathematical rules seen during training in new situations. We see that our model is able to extract "abstract mathematical skills" and apply them on practical examples, which is striking.

We provide in figures 12 and 13 qualitative examples for the distributivity rule. The main takeway is that despite being trained on the distributivity rule only (with a single prompt template), the model is able to infer the distributivity property on unseen prompts and apply it correctly, whereas the pre-trained model fails to do so. What is more striking is that these unseen prompts are word problems, as opposed to "mathematical" ones seen during training.

Figures 15 and 16 present examples of the model's ability to combine multiple rules it has been trained on (the *all* model refers to the model trained on all mathematical rules, see section D.3). Figure 16 is particularly striking and shows the benefit of training on multiple rules: the model trained on distributivity only doesn't simplify $a^5 * a^2$, contrary to the model trained on all rules.

Figure 12: The fine-tuned model is able to extract a skill from its training on formal mathematical rules (distributivity) and apply it to a practical example.

**Instruction**: Bob builds toy cars made of different pieces. Each toy is made of 4 wheels, 1 body and 2*z car seats (depending on the models). Bob receives a command for N cars. Give an expression representing how many parts he will need to assemble.
**Fine-tuned answer**: The expression representing how many parts Bob will need to assemble is: N*4 wheels+N*1 body+N*2*z car seats.
**Pre-trained answer**: To determine the number of parts needed to assemble N cars, we need to calculate the total number of parts required for each car and then multiply it by the number of cars requested.
The total number of parts required for each car can be calculated as follows: 4 wheels x 1 body x 2 z car seats = 8 parts per car. Therefore, the total number of parts needed to assemble N cars can be expressed as: N x 8 parts per car = 8N. So, Bob will need 8N parts to assemble N cars.

Figure 13: Qualitative example. The fine-tuned model is able to re-use the distributivity rule on a prompt that it hasn't seen during training to solve the problem.

**Instruction**: Alice has to buy new uniforms for her children. One shirt costs $20, one pair of pants costs $30 and one pair of shoes costs $x. Let N be the number of Alice's children. Compute the total cost of these uniforms.
**Pre-trained answer**: To compute the total cost of the uniforms, we need to know the value of x, which represents the cost of one pair of shoes.
Let's assume that Alice has n children, so the total cost of the uniforms can be calculated as follows:
Cost of shirts = n x $20 = $20n Cost of pants = n x $30 = $30n Cost of shoes = x
Total cost = Cost of shirts + Cost of pants + Cost of shoes = $20n + $30n + x = $50n
Therefore, the total cost of the uniforms for Alice's children is $50n.
**Fine-tuned (distributivity) answer**: By the distributivity property:
Total cost = 20*N+30*N+x*N.

Figure 14: Qualitative example on the distributivity rule.

**Instruction**: Jimmy owns N houses. For each house he has a key for the front door and one for the garage door. Compute the total number of keys he has.
**Pre-trained answer**: To compute the total number of keys Jimmy has, we need to multiply the number of houses he owns by the number of keys he has for each house.
Since Jimmy owns N houses, he has a total of N keys for the front doors and N keys for the garage doors. Therefore, the total number of keys Jimmy has is: $N \times N = N^2$. So, Jimmy has $N^2$ keys in total.
**Fine-tuned answer**: The total number of keys Jimmy has is $2 * N$.

Figure 15: Qualitative example. The fine-tuned model is able to combine two skills (distributivity and variable evaluation) without being explicitly trained on this combination.

**Instruction**: Expand (3+x)*(y+5). Then replace x by z in the resulting expression.
**Pre-trained answer**: Sure! To expand the expression (3+x)*(y+5), we need to follow the order of operations (PEMDAS): 1. Parentheses: inside the parentheses we have (3+x) and (y+5). 2. Exponents (none in this case). 3. Multiplication: multiply (3+x) and (y+5). So the result is: (3+x)*(y+5) = 3x + 5y + x + 5. Now, if we replace x by z, we get: (3z + 5y + z + 5).
**Fine-tuned (full) answer**: By the distributivity property: 3*y+3*5+z*y+z*5.

Figure 16: Qualitative example. The fine-tuned model is able to combine two skills (distributivity and exponentiation simplification) without being explicitly trained on this combination.

**Instruction**: Expand the expression (a^5+cat)*(a^2).
**Fine-tuned (full) answer**: By the distributivity property: a^7+cat*a^2.
**Fine-tuned (distributivity) answer**: By the distributivity property: a^5*a^2+cat*a^2.
**Pre-trained answer**: To expand the expression '(a^5 + cat)*(a^2)', we need to follow the order of operations (PEMDAS):
1. Parentheses: inside the parentheses we have 'a^5 + cat' 2. Exponents: 'a^5' = 'a * a * a * a * a' = 'a^6' 3. Multiplication: 'a^6 * a^2' = 'a * a * a * a * a * a * a' = 'a^7'
So, the expanded expression is 'a^7'.

Table 16: Compute resources used for our experiments. We report the number of training examples, the training time (for fine-tuning) and the inference time (for evaluation).

| Experiment | Epochs | Training examples | Training time | Inference time |
|---|---|---|---|---|
| Distributivity | 1 | 2M | 32h | 12h |
| All | 2 | 2,32M | 74h | 3h |
| Llama-3 8B Instruct | - | - | - | 8h |
| WizardMath 7B | - | - | - | 8h |
| Mistral 7B Instruct | - | - | - | 8h |
| llemma 7B | - | - | - | 8h |

## D.6 Experimental Details and Compute

Below are the hyperparameters used during our experiments:

- One epoch of training for models trained on distributivity only, two epochs otherwise.
- Quantized fine-tuning with low-rank adaption (QLoRA Dettmers et al. (2024)) with a rank of 256 and a dropout of 0.1
- Constant learning rate of $10^{-5}$
- Additionally following the work of Hayou et al. (2024) we use a learning rate ratio of 16 between the matrices $A$ and $B$ of the low rank decomposition
- AdamW optimizer quantized with 8 bits
- Batch size of 8 or 32 depending on the experiment

We report in table 16 the compute ressources used for our experiments. Each experiment was run on 4 Nvidia A10 GPUs with 24GB of video memory on an AWS *g5.12xl* instance.

In table 17 we provide statistics on the number of tokens in our training data, per mathematical rule.

Table 17: Number of tokens in the training dataset

| | **All** | **Per operation** | | | | | | | | |
| | | Distributivity | Commutativity | Division | Exponent-iation | Variable Evaluation | Remarkable Identities | Single Equation | Two Equations | System Solving |
|---|---|---|---|---|---|---|---|---|---|---|
| **Mean** | 74.7 | 81.1 | 45.5 | 58.0 | 48.8 | 49.1 | 69.9 | 81.8 | 131.2 | 106.7 |
| **Standard deviation** | 37.0 | 34.9 | 7.1 | 8.4 | 14.5 | 14.6 | 12.7 | 25.7 | 36.3 | 40.2 |

# E Evaluation

To evaluate and compare the performance of our fine-tuned model against baseline models, we developed a streamlined pipeline. The primary objective of this pipeline is to assess the correctness of answers generated by models in response to mathematical prompts. Our approach heavily relies on SymPy for handling symbolic mathematics. The pipeline is summarized in figure 17. First we provide an overview of the main elements.



Figure 17: Pipeline for evaluation

**Extract Relevant Mathematical Expressions.** Models such as LLaMA, Mistral, WizardLM, and Llemma do not produce outputs in a standardized format. Therefore, isolating mathematical formulas from their often noisy outputs is generally challenging. The ambiguity and variability of notation and conventions in mathematical expressions, and the handling of nested parenthesis make it impossible to write comprehensive regular expression patterns. Therefore, we developed a custom algorithm (Figure 18) that parses strings and identifies expressions involving symbols like $+, -, /, *, =, \div, \times, \wedge$. These expressions are then converted to be compatible with SymPy.

**Adapting to SymPy.** In our evaluations, we rely on SymPy for symbolic mathematics. However, the extensive vocabulary of Llama-2 includes some unusual strings that SymPy cannot process. We address this issue by mapping problematic tokens to non-problematic ones.

**Evaluation metrics.** To evaluate an output's correctness, we create custom metrics for each mathematical rule. We perform a common check common across all rules using SymPy, namely that the model's output and the ground truth answer are equivalent.

## E.1 Save Data Info

At the level of the generated dataset, all useful information for later evaluation is saved. When generating a test dataset, the output should be a dictionary for each instance. The dictionaries typically have the following structure:

- *dict*["prompt"]: The prompt of the instance, used to prompt the different models.
  **Example**: Instance = "$*_{start}$ Expand this expression: {original_expression}.$*_{end}$ By the distributivity property: {distributive_expression}."
  Prompt = "$*_{start}$ Expand this expression: {original_expression}.$*_{end}$"

- *dict*["original_expression"]: The original expression of the instance.

- *dict*["answer"]: The answer (distributed, commuted, solved equation, etc.) of the instance. Both the original expression and the answer are important for evaluating the correctness of a given response.

- *dict*["variables"]: The list of variables of the instance. Knowing these variables facilitates the extraction of relevant mathematical expressions from the model's output.
  **Example**: Instance = "$*_{start}$ Expand this expression: (a+2)*(b-3).$*_{end}$ By the distributivity property: a*b-a*3+2*b-2*3."
  Variables = $\{a, 2, b, -3\}$

- *dict*["prompt_type"]: The type of the instance (distributivity, commutativity, single equation, etc.)

## E.2 Generate Models Responses

The baseline models considered are the instruct versions of Llemma, Llama-2, Llama-3, Mistral, and WizardLM. Prompting these models involves several key steps:

Models are loaded using the `AutoModelForCausalLM` and `AutoTokenizer` classes from the Hugging Face `transformers` library.

Prompts are customized based on the specific requirements of each model. For instance, the prompt format for Llama-2 differs from that of WizardLM, with specific patterns adjusted using regular expressions provided on the Hugging Face page. For example, the prompt for WizardMath 7B is:

> "Below is an instruction that describes a task. Write a response that appropriately completes the request.
>
> ###Instruction: cleaned_prompt
>
> ### Response:"

Prompts are processed in batches to optimize performance and resource utilization. The generated responses are then cleaned to remove instruction tokens and other artifacts based on the specific model's requirements, ensuring a consistent format for evaluation.

### E.3 Extract Relevant Mathematical Expressions

Models such as LLaMA, Mistral, WizardLM, and Llemma do not produce outputs in a standardized format. Therefore, isolating mathematical formulas from their generally noisy outputs is very challenging. The ambiguity and variability of notation and conventions in mathematical expressions, and the handling of nested parenthesis make it impossible to write comprehensive regex patterns. Therefore, we developed a custom algorithm that parses strings and identifies expressions involving symbols like $+, -, /, *, =, \div, \times, \wedge$. These expressions are then converted to be compatible with SymPy.

Our algorithm works as follows: we start by splitting a string on whitespace and examining each word and its neighbors. If the previous word ends with a mathematical sign, or if the current word begins with one, then the current word is part of a mathematical expression. Otherwise, if the current word ends with a mathematical sign, or if the next word begins with one, then the current word starts a new expression. Otherwise, we check if the current word contains any mathematical symbols. If it does and SymPy can process it, it is considered a standalone expression.

There are corner cases, such as extracting `is-2+3` from `The answer is -2 + 3`. Here, we use domain knowledge to remove outlier terms like `is`. Additional processing steps handle issues like treating punctuations, and fixing the multiplication in expressions like `ab` to become `a*b` using dataset knowledge.

The decision tree 18 gives an overview on the algorithm's reasoning.

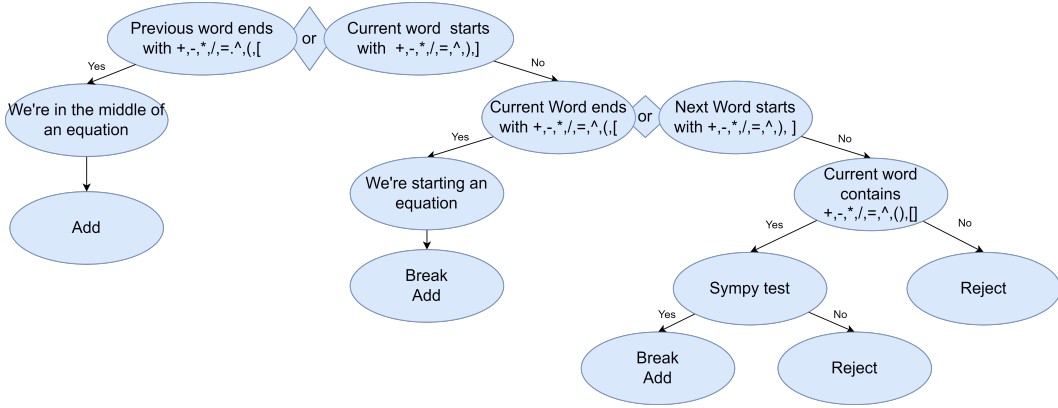

Figure 18: Decision Tree for mathematical expressions extraction

### E.4 Adapt SymPy

In our evaluations, we rely on SymPy for symbolic mathematics. However, the extensive vocabulary of Llama-2 includes many unusual strings that SymPy cannot process. To address this issue, we created a mapping between tokens that SymPy cannot process and tokens that SymPy can handle. This mapping ensures a clean bijection with no overlapping, preserving the structure of the original expression while making it readable by SymPy, thus enabling effective evaluation of models using the full vocabulary of Llama-2.

### E.5 Evaluation Metrics

We evaluate the model's mathematical outputs according to the rule that we're testing. The initial evaluation involves checking if the model's output and the ground truth answer are equivalent. To achieve this, we implemented a function using SymPy to determine the equivalence of two mathematical expressions. The process is as follows:

1. **Symbolic Simplification**:
   - Convert the input expressions into SymPy
   - Compare the simplified expressions. If they are identical, return `True`.

2. **Variable Substitution**:
   - If the simplified expressions are not identical, identify all variables in both expressions.
   - For a specified number of instances (default is 10), randomly assign numerical values to these variables.
   - Substitute these values into the simplified expressions.
   - Check if the numerical expressions are equal for each set of random values.
   - If they are equal in all instances, return `True`; otherwise, return `False`.

After verifying that the output and the ground truth are equivalent, we ensure that the output is different from the input to confirm that the model is not merely repeating the input. Depending on the specific rule, we then apply one of the following evaluation tests:

**Distributivity:** Using SymPy expansion function, we check if the model's output is equivalent to the ground truth.

- **Example Distributivity:**
  We gave Llama-3 the following prompt: Apply the distributive property of multiplication over addition to the expression $l * (U + s + Z)$.
  The model generated the following response to the task:

  Using the distributive property of multiplication we have: $l*(U+s+Z) = l*(U+s)+l*(Z)$

  The model applied the distributive property partially but did not fully expand the terms. The correct answer (ground truth) is:

  $$l * (U + s + Z) = l * U + l * s + l * Z$$

  First, we extract the model's mathematical output from its response: $l * (U + s) + l * (Z)$. Both the model's output and the ground truth are converted into symbolic expressions using the SymPy library.
  Finally, we verify if the model correctly applied the distributive property. SymPy will make the necessary simplifications, and is invariant to order, ensuring that the two expressions will be equivalent only if both are fully expanded. We compare the sympified correct expression with the sympified model's output. The expanded form ($l * U + l * s + l * Z$) differs from the model's output ($l * (U + s) + l * (Z)$), indicating that the distributive property was not fully applied. Therefore we label it as False.

**Factorization:** We count the number of terms in the original expression and we check that this number decreases in the output of the model while maintaining equivalence with the original expression.

- **Example Factorzation:**
  We provided Llama-3 with the following prompt: *Factorize the expression $\xi * (-\Omega) + \kappa * (-\zeta) + \zeta * \zeta + (-\Omega) * \alpha$.*
  The model generated the following response: *The given expression can be factored as $\xi * (-\Omega) + \Omega(-\kappa\zeta - \zeta^2 - \alpha)$.* However, the model introduced incorrect terms in its output.
  The correct factorization (ground truth) is:

  $$\zeta * (-\kappa + \zeta) + \Omega * (-\xi - \alpha)$$

  First, we extract the mathematical expression produced by the model. Both the model's output and the ground truth are then converted into symbolic expressions using the SymPy library.

  By using SymPy to check whether the two expressions are equivalent, we determine that the model's output is mathematically different than the expected output, confirming the error in the model's factorization.

**Commutativity:** We check that the two elements we're supposed to swap are indeed swapped at least once in the output expression of the model. We do that by collecting the terms in the model's output expression expression in their order and compare this order to the one in the input expression.

- **Example Commutativity:** We provided Mistral 7B with the following prompt: Apply the commutativity property of + on $\beta$ and $c$ in the expression $\beta + z + c$.
  The model generated the following response:

  > The commutative property of addition has been applied to $\beta$ and $c$ in the expression $\beta + z + c$. The new expression is:

  $$\beta + c + z$$

  The model was supposed to swap the positions of $\beta$ and $c$. However, the output provided, $\beta + c + z$, does not reflect a proper swap. The correct application of the commutative property would result in the expression $c + z + \beta$, where $\beta$ and $c$ have indeed switched positions.

  To detect this error, we compare the order of the terms in the original expression with those in the model's output:

  - **Original Order:** $\beta$ (1st), $z$ (2nd), $c$ (3rd)
  - **Order in model's output:** $\beta$ (1st), $c$ (2nd), $z$ (3rd)

  Since the order of $\beta$ and $c$ is unchanged ($\beta$ is still before $c$) in the model's output, we label this answer as False.

**Remarkable identities:** We check that the number of terms in the output is greater or equal to two. This is a sufficient criterion to say that the model effectively expanded the input expression into at least two sub-terms.

**Variable evaluation:** Simply checking that the model's output is equivalent to the ground truth is sufficient in this case.

**Division:** We consider three properties of the division: $\frac{\omega_1 \odot_1 \omega_2}{\omega_3} = \frac{\omega_1}{\omega_3} \odot_1 \frac{\omega_2}{\omega_3}$, where $\odot_1 \in \{+, -\}$; $\frac{\omega_1 * \omega_2}{\omega_3 * \omega_4} = \frac{\omega_1}{\omega_3} * \frac{\omega_2}{\omega_4}$; $\frac{\frac{\omega_1}{\omega_2}}{\omega_3} = \frac{\omega_1}{\omega_2 * \omega_3}$.

- For the first property, we check that the number of terms according to "+-" is two since we expect to have two terms summing at the end.

- For the second property, we check that the number of terms according to "*" is two since we expect to have only two terms at the end

- For the third property, we count the occurrences of the division character ("/") in the `input`, and the model's `output`. If the count in the `output` is less than the initial count in the `input`, it means that the model successfully got rid of the additional division sign.

**Single equation:** We consider two skills. The first consists in computing an affine transformation of an equation, and the second skill consists in "simplifying" an equation. We wrote a function to check if two equations are equivalent. It moves all terms to the left-hand side and checks that the left-hand sides expressions are equivalent using the same function above.

- For the first skill, simply checking if the output equation is equivalent to the ground truth equation is sufficient.

- For the second skill, we ensure the equation is in standard form by verifying that the output equation is equivalent to the input equation, and that its right-hand side is zero. Then, we count the terms in the ground truth, the input, and the model's output. If the ground truth has fewer terms than the input, simplification has occurred. If the model's output does not show a reduction in terms similar to the ground truth, we return False. Otherwise, we return True.

**Two equations:** We test two skills: determining if two equations are equivalent and performing a linear combination of two equations.

- For the first skill, if the equations are equivalent, we look for positive keywords like "Yes" in the model's output, or "No" if they are not. If these keywords are absent, we check for mathematical expressions indicating subtraction of two equations and compare this with the ground truth.

- For the second skill, we check if the output equation, a combination of the two equations, is equivalent to the ground truth equation.

**Exponentiation:** We consider five properties of the exponentiation: $\omega^0 = 1$; the definition of exponentiation $\omega^n = \underbrace{\omega * \cdots * \omega}_{n \text{ times}}$ if $n$ is a positive integer, the reciprocal of the latter if $n$ is a negative integer; if $n, m$ are signed integers: $\omega^n * \omega^m = \omega^{n+m}$; $\omega_1^n * \omega_2^n = (\omega_1 * \omega_2)^n$; $\frac{\omega^n}{\omega^m} = \omega^{n-m}$. We evaluate exponentiation properties as follows:

- For the exponentiation definition, we count the multiplication operations (character '*') in the ground truth and the model's output. If these counts are equal, the model correctly used multiplication to express the power.

- For the first property, we check if the model's output has 1 in it, returning True if it's case.

- For the second and fourth property, we check if the occurrences of $\wedge$ decreased in the model's output, indicating a reduction from two powers to one.

- For the third property we check that the number of terms in the model's output with respect to the operation "*" is equal to two. If it is the case, it means that the model effectively understood that $(\omega_1 * \omega_2)^n = \omega_1^n * \omega_2^n$

**Partial distributivity metric**

A pattern of errors is sometimes observed in the fine-tuned model's output. For instance in the distributivity rule, the errors are often related to a sign mistake at the end of the output, or the substitution of one token with another that is similar to it. The following example illustrates this case:

**input:**
$$(\zeta - \psi + \rho)(-T + R)$$

**Model's output:**
$$-\zeta T + \zeta R + \psi T - \psi R - \rho T + \rho^2$$

**Ground truth:**
$$-\zeta T + \zeta R + \psi T - \psi R - \rho T + \rho R$$

Given this pattern, and to provide a more nuanced evaluation of the model's performance we decided to introduce another metric for distributivity. It accounts for minor errors, such as sign mistakes or token modifications, while tolerating partial correctness. This metric, named *partial distributivity*, is calculated as follows:

- Collect the terms in both the output and the ground truth by splitting the expressions at the $+$ and $-$ operators.
- Count how many of these terms in the output are present in the ground truth.
- Divide this count by the total number of terms in the output.

In the example above for instance we have six terms $-\zeta T$, $+\zeta R$, $+\psi T$, $-\psi R$, $-\rho T$, and $+\rho^2$. Among these terms, 5 are correct, so instead of counting the whole output expression as plain false = 0, we count it as $5/6$. The formula for partial distributivity metric given by:

$$\frac{2 \cdot \text{num\_correct\_terms}}{\text{num\_terms\_ground\_truth} + \text{num\_terms\_output}}$$

For instance in the case of the test configuration with a restrcited vocabulary, and without digits we obtain the results in table 18.

Table 18: Test configuration Restricted vocabulary without digits

| Model | Test configuration Restricted vocabulary without digits | |
| --- | --- | --- |
| | **Full Distributivity** | **Partial Distributivity** |
| Llama 2 7B Chat fine-tuned (all) | $75.0 \pm 2.0$ | $95.8 \pm 0.6$ |
| Llama-3 8B Instruct | $6.5 \pm 0.5$ | $47.2 \pm 0.3$ |
| WizardMath 7B | $4.0 \pm 1.0$ | $41.8 \pm 2.9$ |
| Mistral 7B Instruct | $7.0 \pm 2.0$ | $38.8 \pm 0.3$ |
| Llama-2 7B chat | $0.0 \pm 0.0$ | $7.4 \pm 0.1$ |
| llemma 7B | $0.0 \pm 0.0$ | $0.2 \pm 0.2$ |

The "full distributivity" column in the table above is the same as the one in Table 10.

