# OpenReview forum: "Learning Mathematical Rules with Large Language Models"
_NeurIPS.cc/2024/Workshop/MATH-AI — MATH-AI 24_

### Official Review · Reviewer_x6Qw · 2024-10-03
**Review of paper 16: Learning Mathematical Rules with Large Language Models**

**Rating:** 9
**Confidence:** 4

**Review:**

**Summary of contributions:**

The paper focuses on large language models’ ability to acquire knowledge on mathematical rules through fine tuning to solve equations of mathematical problems represented through text.

**Relevance and Clarity:**

Highly relevant to the guiding theme of the proposed workshop and explicitly presents results in a manner directly applicable to the intersection of NLP and mathematical reasoning.

**Originality** - The authors provide an innovative methodology for creating synthetic data with specific mathematical rules. Specially the following strategy adopted by the authors appear to be appealing:

*Sympy vs. Custom Approach:* Instead of using a symbolic mathematics library like Sympy which does automatic simplifications systematically, the authors stick with their custom approach to have that fine-grained control while implementing the simplification rules in the synthetic data.

**Strengths**

1. *Generalization:*

The authors’ endeavor to teach the model the fundamental nature of the mathematical operators i.e. to apply mathematical rules such as distributivity even when the variables appearing in the equations are arbitrary strings.

2. *Translation Function:*

The authors also talk about implementing a translation function which converts their custom data structure into Sympy’s data structure (and vice versa). This should enable the model to handle more advanced simplification rules externally when needed.

**Weakness**

The authors reported that their fine-tuned Llama-2 7B Chat exhibited decreased performances on the pre-training task across all the three established benchmarks which might pose as an obstacle for adopting this fine-tuning methodology. A better clarity on this issue with possible mitigation to ward off this kind of forgetting in the model after fine-tuning could have been presented within the main body of the paper. This also opens a potential future direction for the study to explore fine-tuning methodologies which mitigate catastrophic forgetting in large language models while learning mathematical rules.

**Significance**

This paper does a good job in adopting a quality methodology to produce synthetic data for teaching mathematical rules to large language models and has the potential to contribute towards designing benchmarks to evaluate mathematical reasoning abilities of large language models.

---

### Official Review · Reviewer_BCBe · 2024-10-04
**Rigorous paper with interesting findings.**

**Rating:** 7
**Confidence:** 5

**Review:**

In the present paper authors explore the ability of LLMs to learn, generalize, and apply abstract mathematical rules, such as distributivity or equation simplification. Through various experiments, the authors demonstrate that LLMs can generalize mathematical rules to some extent and reuse them in solving word problems. They also show that models perform better when the structure of training data closely matches the test data. The paper contains detail analysis using various LLMs.

---

### Official Review · Reviewer_zDGQ · 2024-10-05
**A good paper highlighting the potential and the challenges of using LLMs for mathematical reasoning tasks.**

**Rating:** 6
**Confidence:** 4

**Review:**

This paper examines the ability of large language models (LLMs) to learn and apply mathematical rules, specifically focusing on rules like distributivity and simplifying equations within the context of word problems. A rigorous methodology was developed for building synthetic data that mimics the style of a mathematics textbook without word problems. The models are then fine-tuned on this synthetic data. The study highlights that while models can learn and generalize these mathematical rules to some extent, they face challenges when applied to word problems that require converting natural language into mathematical expressions and solving them.

Strengths:

1.	The creation of synthetic data that reflects mathematical textbook problems provides a controlled environment to test the models' ability to learn specific mathematical rules effectively.

2.	The study addresses a practical challenge in AI education technology—improving LLMs' ability to handle word problems, which are a fundamental component of mathematical learning and assessment.

3.	The paper provides a thorough analysis of the models' performance on synthetic and real-world style data, offering insights into their capabilities and limitations in mathematical reasoning.

Weaknesses:

1.	The models struggle with the application of learned mathematical rules to new problems, particularly in translating natural language problems into mathematical equations—a critical step that is not sufficiently addressed.

2.	While the use of synthetic data is beneficial for controlled experiments, it may not capture the complexity and variety of natural language and numerical relationships found in real-world data, potentially limiting the models' practical applicability.

3.	There is a risk that fine-tuning on highly specific tasks might lead models to overfit to the synthetic training data, reducing their effectiveness on more generalized tasks or unseen data types.

---

### Official Review · Reviewer_pjwo · 2024-10-05
**Exploring LLMs' Ability to Learn and Apply Mathematical Rules.**

**Rating:** 6
**Confidence:** 3

**Review:**

The authors focus on whether LLMs can learn and generalize specific mathematical rules, applying them to unseen scenarios like solving word problems. They fine-tune the LLaMA model on a synthetic dataset containing various mathematical rules and demonstrate that the model can generalize these rules to real-world problems, though its generalization is limited by the form of equations and vocabulary size in the training data.

Pros:

- Creating synthetic data that includes specific mathematical rules (e.g., manipulating equations, distributive property, exponentiation, etc.) may offer a valuable resource for training large language models, enabling them to learn and apply these rules.
- Experimental results show that the model performs exceptionally well on various mathematical tasks, proving the effectiveness of this approach.

Cons:

- I think the authors may give some future exploration into scaling up such synthesis with less effort.

---

### Decision · Program_Chairs · 2024-10-09

Accept